# Uncertainty analysis of total ozone derived from direct solar irradiance spectra in the presence of unknown spectral deviations

Anna Vaskuri[1], Petri Kärhä[1], Luca Egli[2], Julian Gröbner[2], and Erkki Ikonen[1, 3]

[1]Metrology Research Institute, Aalto University, PO Box 15500, 00076 Aalto, Finland
[2]Physikalisch-Meteorologisches Observatorium Davos, World Radiation Center, Dorfstrasse 33, 7260 Davos Dorf, Switzerland
[3]MIKES Metrology, VTT Technical Research Centre of Finland Ltd, PO Box 1000, 02044 VTT, Finland

*Correspondence to:* Anna Vaskuri (anna.vaskuri@aalto.fi), Petri Kärhä (petri.karha@aalto.fi)

**Abstract.** We demonstrate a Monte Carlo model to estimate the uncertainties of total ozone column (TOC), derived from ground-based direct solar spectral irradiance measurements. The model estimates the effects of possible systematic spectral deviations in the solar irradiance spectra on the uncertainties in TOC retrieved. The model is tested with spectral data measured with three different spectroradiometers at an intercomparison campaign of the research project "Traceability for atmospheric total column ozone" at Izaña, Tenerife on 17 September 2016. The TOC values derived at local noon have expanded uncertainties of 1.3% (3.6 DU) for a high-end scanning spectroradiometer, 1.5% (4.4 DU) for a high-end array spectroradiometer, and 4.7% (13.3 DU) for a roughly adopted instrument based on commercially available components and an array spectroradiometer when correlations are taken into account. Neglecting the effects of systematic spectral deviations, the uncertainties reduce by a factor of 3. The TOC results of all devices have good agreement with each other, within the uncertainties, and with the reference values of the order of 282 DU during the analysed day, measured with Brewer spectrophotometer #183.

## 1 Introduction

Atmospheric ozone has been defined as an essential climate variable in the global climate observing system (GCOS-200 (2016)) of the World Meteorological Organization (WMO). Its long-term monitoring is necessary to document the expected recovery of the ozone layer due to the implementation of the Montreal protocol (UNTC (1987)) and its amendments on the protection of the ozone layer. Atmospheric ozone, first discovered by Fabry and Buisson (1913), protects the humans, the biosphere, and infrastructures from adverse effects of ultraviolet (UV) radiation by shielding the Earth surface from excessive radiation levels (McElroy and Fogal (2008)). Since the 1970's, it is known that human-produced chlorofluorocarbons (CFCs) destruct atmospheric ozone (Molina and Rowland (1974)) and have led to recurring massive losses of total ozone in the Antarctic in the form of the ozone hole (Farman *et al.* (1985); Solomon *et al.* (1986)). An unprecedented ozone depletion has also been recently observed in the Arctic (Manney *et al.* (2011)), while in the middle-latitudes, moderate ozone depletion has been observed (Solomon (1999)). The Montreal protocol and its amendments have been successful in reducing the emission of ozone-depleting substances (Velders *et al.* (2007)). Nevertheless, recent studies give conflicting results with respect to the

observation of a recovery of the ozone layer, and model projections have shown the recovery to occur not before the middle of the 21st century (Ball *et al.* (2018); Weber *et al.* (2018)). Therefore, careful monitoring of the thickness of the ozone layer with uncertainties of 1% or less is crucial in verifying the successful implementation of the Montreal Protocol and the eventual recovery of the ozone layer to pre-1970's levels.

"Traceability for atmospheric total column ozone" (ATMOZ) was a three-year project funded partly by the European Metrology Research Programme (EMRP) and the European Union (ATMOZ project (2014 – 2017)). The goal of this project was to produce traceable measurements of total ozone column (TOC) with uncertainties down to 1%, by a systematic investigation of the radiometric and spectroscopic aspects of the methodologies used in retrieval. Another objective of the project was to provide a comprehensive treatment of uncertainties of all parameters affecting the TOC retrievals using

spectrophotometers. This paper presents outcome of the work on studying the uncertainty of TOC obtained from spectral direct solar irradiance measurements, taking unknown spectral errors explicitly into account.

TOC can be determined from spectral measurements of direct solar UV irradiance (Huber *et al.* (1995)). We have developed a Monte Carlo (MC) based model to estimate the uncertainties of the derived TOC values. One frequently overlooked problem with uncertainty evaluation is that the spectral data may hide systematic wavelength dependent errors

due to unknown correlations (Kärhä *et al.* (2017b, 2018); Gardiner *et al.* (1993)). Omitting possible correlations may lead into underestimated uncertainties for derived quantities, since spectrally varying systematic errors typically produce larger deviations than uncorrelated noise-like variations that traditional uncertainty estimations predict. Complete uncertainty budgets for quantities measured are necessary to understand long term environmental trends, such as changes in the stratospheric ozone concentration (e.g. Molina and Rowland (1974)) and solar UV radiation (e.g. Kerr and McElroy (1993); McKenzie *et al.*

20    (2007)).

Physically, spectral correlations may originate, e.g., from lamps or other light sources used in calibrations. If their temperatures change e.g. due to ageing or current setting, a spectral change in the form of Planck's radiation law is introduced. Non-linearity in the responsivity of a detector causes systematic differences between high and low measured values. The introduced spectrally systematic but unknown changes in irradiance may change the derived TOC values significantly,

exceeding the uncertainties calculated assuming that the uncertainty in irradiance behaves like noise. The presence of correlations in measurements can be seen in many ways. For example, problems have occurred when new ozone absorption cross-sections have been taken into use (Redondas *et al.* (2014); Fragkos *et al.* (2015)). Derived ozone values may change significantly because different systematic errors are included in the different cross-sections. Also, TOC estimated from a measured spectrum often depends on the wavelength region chosen, although the measurement region should not affect the

result much.

In this paper, we introduce a new method for dealing with possible correlations in spectral irradiance data and analyse uncertainties in ozone retrievals for three different spectroradiometers used in a recent ATMOZ intercomparison campaign at Izaña, Tenerife, to demonstrate how it can be used in practice. One of the instruments is QASUME (Gröbner *et al.* (2005)) that is the World reference UV spectroradiometer at the World Radiation Center (PMOD/WRC). The second one is an array-based

high-quality spectroradiometer BTS2048-UV-S-WP (BTS) from Gigahertz-Optik (Zuber *et al.* (2017a, b)), operated by

Physikalisch-Technische Bundesanstalt (PTB). The third one is an array-based spectroradiometer AvaSpec-ULS2048LTEC (AVODOR) from Avantes, operated by PMOD/WRC. The field of view of the spectroradiometers has been limited so that they measure direct spectral irradiance of the Sun, excluding most of the indirect radiation from the remainder of the sky.

## 2 ATMOZ field measurement campaign and instrument description

The ATMOZ project arranged a field measurement campaign (ATMOZ campaign (2016)) that took place 12 – 30 September 2016 at the Izaña Atmospheric Observatory, a high mountain Global Atmospheric Watch (GAW) station located at an altitude of 2.36 km above the sea level in Tenerife, Canary Islands, Spain (28.3090$^\mathrm{o}$ N, 16.4990$^\mathrm{o}$ W). The measurement campaign was organised by the Spanish Meteorological Agency (AEMET) and the PMOD/WRC for the intercomparison of TOC measured with different participating instruments, including Dobson and Brewer spectrophotometers, and various spectroradiometers.

The focus of this paper is to study uncertainties of the TOC values derived from direct solar UV irradiance spectra. Total ozone column TOC is the vertical ozone profile $\rho_{\mathrm{O_3}}(z)$ integrated over altitudes as

$$\mathrm{TOC} = \int_{z_0}^{z_1} \rho_{\mathrm{O_3}}(z)\,\mathrm{d}z \tag{1}$$

from the station altitude $z_0$ to the top-of-the-atmosphere altitude $z_1$. We study the data measured during the day of 17 September 2016 with three different spectroradiometers using the ozone retrieval algorithm introduced in Section 3. Station pressure

was monitored during the campaign and determined to be 772.8 hPa with a standard uncertainty of 1.3 hPa. The ozone and temperature profiles were measured with a sonde during the campaign and examples of them are shown in Fig. 1.

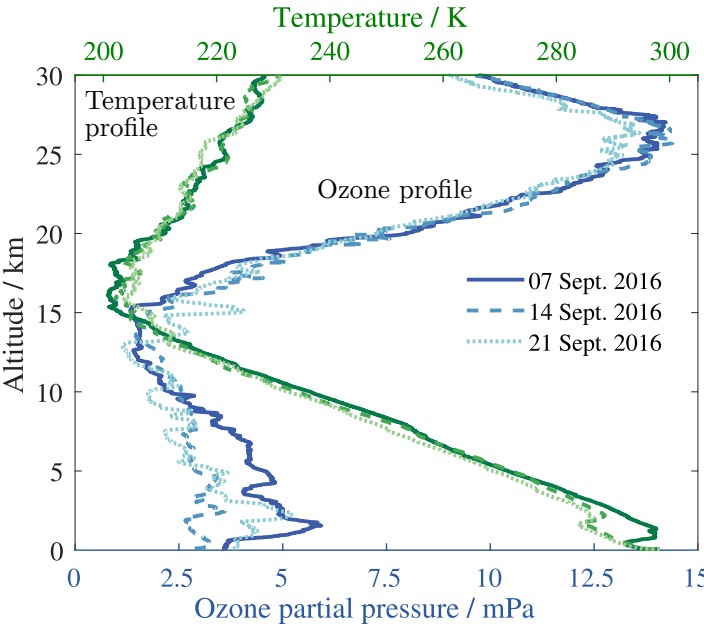

**Figure 1.** Temperature and ozone profiles as a function of the altitude, measured with a sonde during the ATMOZ field measurement campaign.

Our ozone retrieval method uses one atmospheric layer to reduce computational complexity. With the one layer model, the ozone absorption cross-section is a function of the effective temperature and the relative air mass is a function of the effective altitude of the ozone layer. Using the vertical ozone profile $\rho_{O_3}(z)$, the effective altitude $h_{\text{eff}} = 26 \text{ km} \pm 0.5 \text{ km}$ of the ozone layer was estimated by integration over altitudes

$$h_{\text{eff}} = \frac{\int_{z_0}^{z_1} z \, \rho_{O_3}(z) \, dz}{\int_{z_0}^{z_1} \rho_{O_3}(z) \, dz}, \tag{2}$$

from the station altitude $z_0$ to the top-of-the-atmosphere altitude $z_1$. The corresponding effective temperature $T_{\text{eff}} = 228 \text{ K} \pm 1 \text{ K}$ was estimated (Komhyr *et al.* (1993); Fragkos *et al.* (2015)) as

$$T_{\text{eff}} = \frac{\int_{z_0}^{z_1} T(z) \, \rho_{O_3}(z) \, dz}{\int_{z_0}^{z_1} \rho_{O_3}(z) \, dz}. \tag{3}$$

The uncertainties stated for $h_{\text{eff}} = 26 \text{ km} \pm 0.5 \text{ km}$ and $T_{\text{eff}} = 228 \text{ K} \pm 1 \text{ K}$ are standard deviations estimated from the vertical profiles in Fig. 1, measured during the campaign.

The data sets measured by three different spectroradiometers were studied in this work. These spectroradiometers use different techniques to measure the spectral distribution of radiation. Monochromator-based spectroradiometers, such as QASUME, measure one nearly monochromatic wavelength band at a time, and thus measuring the full spectrum is relatively slow. On the other hand, they usually have significantly better stray light properties than array-based spectroradiometers, such as BTS and AVODOR that image the full spectrum at once by dispersing the incoming radiation towards a photodiode array.

QASUME spectroradiometer collects and guides the incoming radiation with input optics and a quartz fiber bundle to the entrance slit of a Bentham DM150 double monochromator (Gröbner *et al.* (2005)). One wavelength at a time is selected by rotating the two gratings of the double-monochmomator. Then, the monochromatic signal is measured with a photomultiplier tube. QASUME is usually operated in global spectral irradiance mode (Gröbner *et al.* (2005); Hülsen *et al.* (2016)), but during the campaign it was equipped with a collimator tube with a full opening angle of 2.5° allowing the measurement of direct solar spectral irradiance (Gröbner *et al.* (2017)). The measurement range of QASUME during the campaign was limited to 250 nm – 500 nm with a step interval of 0.25 nm, so that one spectrum was measured every 15 minutes. To ensure stable outdoor measurements, the double-monochromator of QASUME was mounted inside a temperature-controlled weather-proofed box (Hülsen *et al.* (2016)).

BTS spectroradiometer utilizes a stationary grating and a back-thinned cooled CCD array detector, mounted in a Czerny-Turner configuration (Zuber *et al.* (2017a, b)). To measure direct solar spectrum, BTS was equipped with a collimator tube with a full opening angle of 2.8° designed by PTB, and it uses an internal filter wheel system with eight filter positions together with a specific measurement routine to reduce stray light. BTS was mounted on a solar tracker, EKO STR-32G by EKO Instruments Co., Ltd., with pointing accuracy better than 0.01°. A weather-proof housing with temperature control allows BTS operation at the ambient temperatures from –25 °C to 50 °C. During the ATMOZ campaign, the housing temperature of BTS was measured to be stable within 0.1°C (Zuber *et al.* (2017a, b)). The measurement range of BTS was 200 nm – 430 nm with a step size of 0.2 nm during the campaign. One spectrum was measured every 45 seconds.

AVODOR spectroradiometer has a stationary grating and a back-thinned cooled CCD array detector in a Czerny-Turner configuration. AVODOR measures the spectrum from 200 nm to 540 nm with a step size of 0.14 nm in the UV region. During the ATMOZ campaign, the field of view of AVODOR was limited to 1.5° by a commercial collimator tube used, J1004-SMA by CMS Ing.Dr.Schreder GmbH. The spectral range of AVODOR was limited between 295 nm and 345 nm by a combination of two solar blind filters to reduce stray light from the visible and infrared parts of the solar spectrum. The solar blind filters were mounted between the collimator tube and the fiber entrance of the spectroradiometer. One spectrum was measured every 30 seconds.

The slit functions of the three spectroradiometers shown in Fig. 2 were measured with lasers before the measurement campaign. They are needed when fitting the modelled spectra at the Earth surface to the measured spectra. In addition, it is of importance to notice the different wavelength steps of the data, 0.25 nm for QASUME, 0.2 nm for BTS, and 0.14 nm for AVODOR. The wavelength steps of the spectral data affect the magnitude of the uncertainties in TOC created by spectrally random components. In our full spectrum TOC retrieval, the number of data points $n$ which is smaller with a larger wavelength step interval, affects uncertainties with a factor of $1/\sqrt{n}$ as demonstrated in (Kärhä *et al.* (2017b); Poikonen *et al.* (2009)).

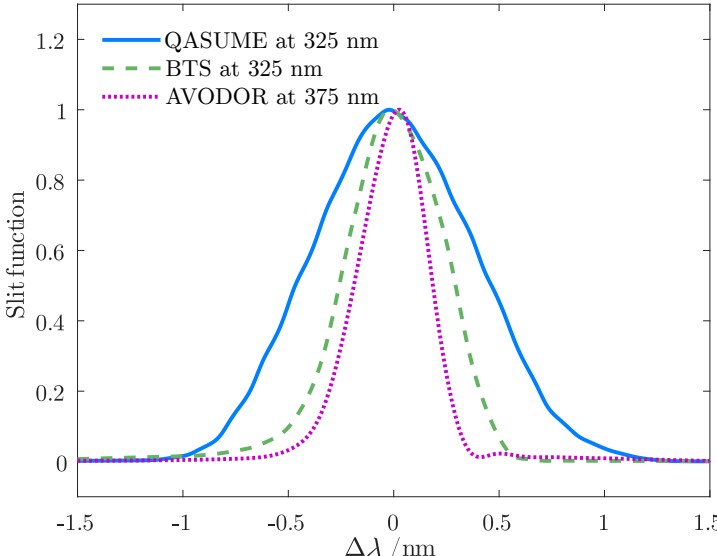

**Figure 2.** Slit functions, measured with narrow band lasers, for the spectroradiometers used in the Izaña campaign. The laser wavelengths are stated in legends.

Brewer MkIII spectrophotometers used as reference devices for ozone measurements established by the International Meteorological Organization, the Commission for Instruments and Methods of Observation (CIMO) (Redondas *et al.* (2016)), also measure the spectral irradiance at UV region, but using four narrow wavelength bands at 310.1 nm, 313.5 nm, 316.8 nm, and 320.0 nm (Kipp & Zonen (2015)). The Brewer MkIII instruments solve absolute TOC values by comparing the logarithms
of ratios of count rates between four wavelength channels, i.e. using the double ratio technique. Determining the TOC using the double ratio method is invariant for such spectral deviations which have the same relative magnitude at all wavelengths, i.e spectrally constant deviations.

Our full spectrum retrieval method performs averaging in the wavelength domain, whereas Brewer spectrophotometer does it in the time domain. Brewer can measure up to tens of seconds to get millions of photons, so that the photon noise reduces
to a level of 0.1%. At this low noise level, it is not critical that only four wavelengths are used. As Brewer instruments are well-known and widely used, we also compare TOC results obtained using our full spectrum retrieval method and the spectroradiometers to those measured by Brewer #183 during the same day.

## 3 Atmospheric model

In this study, we use an atmospheric ozone retrieval algorithm in many aspects similar to the article by Huber *et al.* (1995).
The relationship between the spectral irradiance $E_{\mathrm{s}}(\lambda)$ at the Earth surface and the extra-terrestrial solar spectrum $E_{\mathrm{ext}}(\lambda)$ is

based on Beer–Lambert–Bouguer absorption law (Beer (1852); Lambert (1760); Bouguer (1729)) as

$$E_{\mathrm{s}}(\lambda) = c \cdot E_{\mathrm{ext}}(\lambda) \cdot \exp\left[-\alpha_{\mathrm{O_3}}(\lambda, T_{\mathrm{eff}}) \cdot \mathrm{TOC} \cdot m_{\mathrm{TOC}} - \tau_{\mathrm{R}}(\lambda, P_0, z_0, \phi) \cdot m_{\mathrm{R}} - \tau_{\mathrm{AOD}}(\lambda) \cdot m_{\mathrm{AOD}}\right], \tag{4}$$

where $\alpha_{\mathrm{O_3}}(\lambda, T_{\mathrm{eff}})$ is the ozone absorption cross-section at the effective ozone temperature $T_{\mathrm{eff}}$, $\tau_{\mathrm{R}}(\lambda, P_0, z_0, \phi)$ is the Rayleigh scattering optical depth that depends on the station pressure $P_0$, the station altitude $z_0$, and the geographic latitude $\phi$. The QASUME-FTS data set by Gröbner *et al.* (2017) was used as the extra-terrestrial spectrum $E_{\mathrm{ext}}(\lambda)$. Parameter $c$ is a scaling factor set as a free parameter to compensate for spectrally constant deviations.

The relative air mass of the ozone layer with the Earth curvature taken into account can be expressed as

$$m_{\mathrm{TOC}} = \frac{1}{\cos\left[\arcsin\left(\frac{R}{R + h_{\mathrm{eff}}} \cdot \sin\theta\right)\right]}, \tag{5}$$

where $\theta$ is the incident solar zenith angle at the observing site that is a function of the local time, date, and geographic coordinates (Meeus (1998); Reda and Andreas (2008)). Parameter $h_{\mathrm{eff}}$ is the altitude of the ozone layer from the ground, and $R$ is the radius of the Earth. As the ozone and other molecules creating scattering are distributed at different altitudes, we calculate the relative air mass factor $m_{\mathrm{R}}$ for Rayleigh scattering at the altitude of 5 km (Gröbner and Kerr (2001)) and approximate the relative air mass factor of aerosols so that $m_{\mathrm{AOD}} \approx m_{\mathrm{R}}$ (Gröbner *et al.* (2017)). The temperature dependence of $\alpha_{\mathrm{O_3}}(\lambda, T_{\mathrm{eff}})$ between 203 K and 253 K (Weber *et al.* (2016)) was interpolated by a second degree polynomial at each wavelength (Paur and Bass (1985)) as

$$\alpha_{\mathrm{O_3}}(\lambda, T_{\mathrm{eff}}) = a_1(\lambda) T_{\mathrm{eff}}^2 + a_2(\lambda) T_{\mathrm{eff}} + a_3(\lambda). \tag{6}$$

We take the Rayleigh scattering optical depth into account by the state-of-the-art model by (Bodhaine *et al.* (1999)). The aerosol optical depth (AOD) is approximated from the Ångström AOD model (Ångström (1964)) as

$$\tau_{\mathrm{AOD}}(\lambda) = \beta \cdot \left(\frac{\lambda}{1\,\mu\mathrm{m}}\right)^{-\alpha}, \tag{7}$$

where constant $\alpha \approx 1.4$ is the Ångström coefficient at typical atmospheric conditions and $\beta \geq 0$ is the Ångström turbidity coefficient.

The model spectrum $E_{\mathrm{s}}(\lambda)$ at the Earth surface, convolved by the slit function of the spectroradiometer as shown in Fig. 2, is fitted with parameters TOC, $\beta$, and $c$ to the measured ground-based spectrum $E(\lambda)$. The absolute TOC level obtained depends on the fitting method used. We used a least squares fitting method (Levenberg (1944); Marquardt (1963)) with trust-region optimisation by Matlab function 'lsqnonlin' as

$$S = \sum_{i=1}^{n} w(\lambda_i) \cdot \left[E_{\mathrm{s}}(\lambda_i) - E(\lambda_i)\right]^2, \tag{8}$$

where $S$ is the sum of the squared residuals to be minimised, and $w(\lambda_i)$ is the weight for each point measured. Index $i = 1, 2, ..., n$ runs over the wavelengths of the spectral measurements.

Figure 3 presents examples of measured and modelled spectra for the spectroradiometers used in this work. As can be seen, the signal-to-noise ratios and stray light properties of the devices differ significantly among different spectroradiometers. All spectra measured by QASUME are practically noiseless above $10^{-6}$ Wm$^{-2}$nm$^{-1}$, resulting in a dynamic range of approximately four orders of magnitude. The dynamic range for BTS is approximately two orders of magnitude and less than two orders of magnitude for AVODOR.

For QASUME, we use the relative least squares fitting method (RLS) minimisation with $w(\lambda) = E(\lambda)^{-2}$, as QASUME does not suffer from stray light and RLS fitting has been used in the past for monochromator-based spectroradiometers, e.g. by Huber *et al.* (1995). These least squares fitting selections are discussed in more detail in Appendix A.

To minimise the effect of stray light, we use absolute least squares minimisation, also known as ordinary least squares fitting method (OLS), with $w(\lambda) = 1$ for BTS and AVODOR, as this selection is less affected by the lowest irradiance levels where the stray light and noise are dominant. As shown in Appendix A, using OLS introduces an offset, approximately 1% in these measurements, to the retrieved TOC values. We take this into account by analysing the results at noon that are less influenced by stray light also with RLS, and make a correction to the OLS results as

$$\text{TOC}_{\text{UTC,OLS,c}} = \frac{\text{TOC}_{\text{noon,RLS}}}{\text{TOC}_{\text{noon,OLS}}} \cdot \text{TOC}_{\text{UTC,OLS}}. \tag{9}$$

After this correction, results of all instruments, QASUME, BTS, and AVODOR are comparable.

The shortest fitting wavelength for the spectroradiometers in this work was selected to be 300 nm since the typical stray light compensation methods are not effective below 300 nm (Nevas *et al.* (2014)). The upper fitting wavelength limit was set to 340 nm with all three spectroradiometers as the ozone absorption is not effective above that wavelength. Due to the relatively large bandwidths of the spectroradiometers (Fig. 2), calculations before the convolution and the convolution itself were carried out over a wider range 295 nm – 345 nm.

To see whether a global optimum is achieved with our atmospheric ozone retrieval method, we varied the initial guess values of TOC from 10 DU to 700 DU, $\beta$ from 0 to 0.5, and $c$ from 0 to 100. Within the ranges stated, the free parameters always converged to the same final values regardless of the initial guess values.

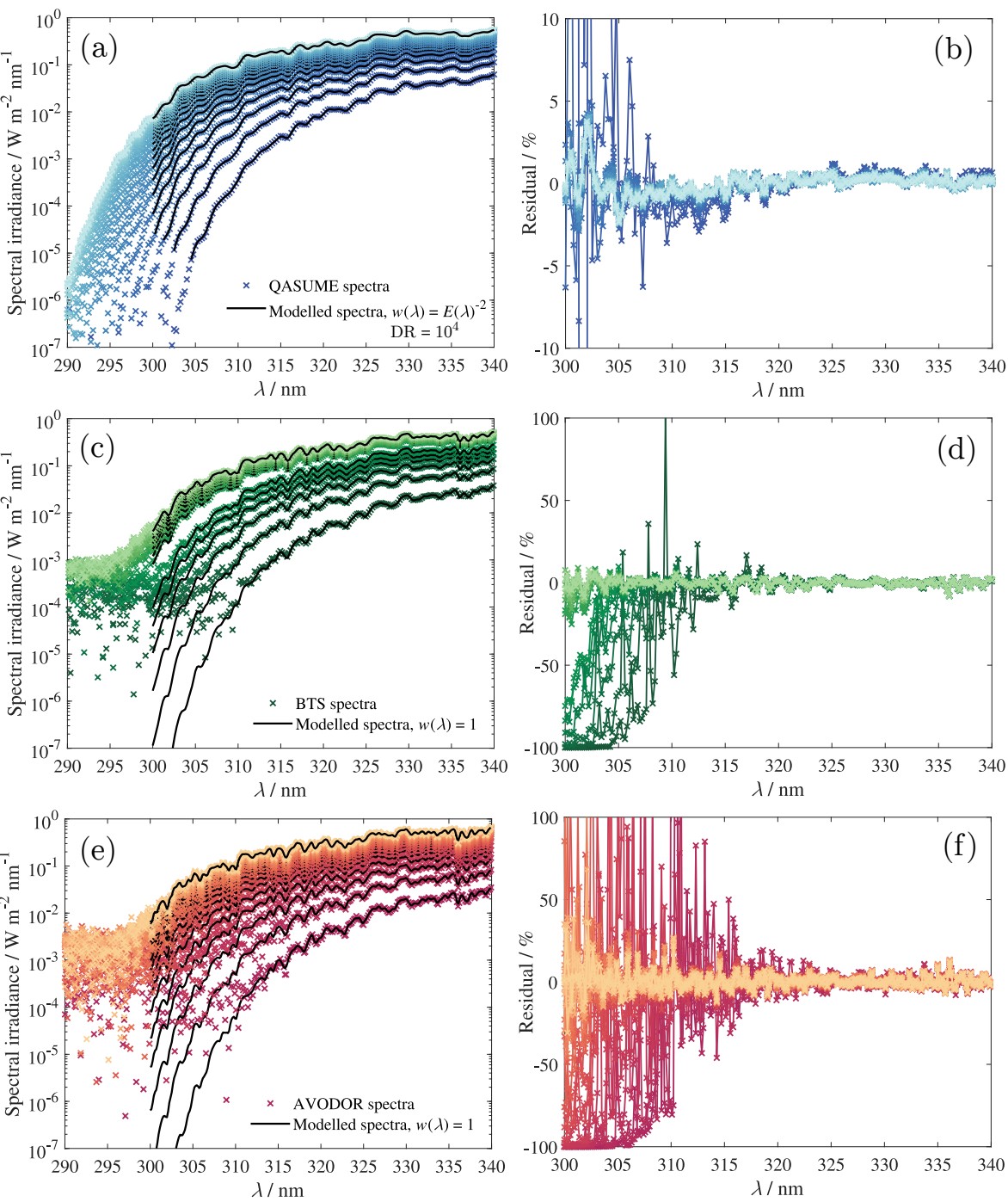

**Figure 3.** Examples of fitting the atmospheric model to the direct ground-based solar UV spectra between 300 nm and 340 nm for QASUME (a–b) , BTS (c–d) and AVODOR (e–f). In figures on the left hand side, the coloured symbols indicate measured spectra, and the black solid curves indicate modelled spectra. Figures on the right hand side show the relative spectral residuals of the fits. In (a), the abbreviation DR refers to the dynamic range of QASUME data used in the least squares fitting.

## 4   Uncertainty estimation

### 4.1   Uncertainty model

In uncertainty analysis, the combined uncertainties are calculated with the square sum of the standard deviations of the components, i.e. their variances are summed up. If correlations of uncertainties are known, they should be taken into account. This can be carried out with the methods defined in the Guide to the Expression of Uncertainty in Measurement (JCGM 100 (2008)). In this paper, we do this for all uncertainty components, where the mechanism of contributing to the uncertainty of TOC is known. However, with some of the components, we do not know exactly the mechanisms leading into correlations. With such uncertainties, we estimate the effects that possible correlations might have using a newly developed MC model described in (Kärhä *et al.* (2017a, b)).

In our MC model, possible systematic deviations within uncertainties are reproduced using a cumulative Fourier series

$$\delta(\lambda) = \sum_{i=0}^{N} \gamma_i f_i(\lambda) \tag{10}$$

with sinusoidal base functions, shown in Fig. 4, as

$$f_i(\lambda) = \sqrt{2} \cdot \sin\left[ i\left( 2\pi \frac{\lambda - \lambda_1}{\lambda_2 - \lambda_1} \right) + \phi_i \right], \tag{11}$$

where index $i = 1, 2, ..., N$ depicts the order of complexity of the deviation (Kärhä *et al.* (2017b)); $\lambda_1$ and $\lambda_2$ limit the wavelength range of the analysis. For calculations before the convolution due to slit function, this range was set to 295 nm – 345 nm. Otherwise, the results at the ends of the range 300 nm – 340 nm would be distorted due to incomplete convolution. This concerns e.g. uncertainty of the extra-terrestrial spectrum and the ozone absorption cross-section. After the convolution, actual fitting of the modelled spectra to the measured spectra was carried out at 300 nm – 340 nm, and this range was also used in the uncertainty analysis of the components related to the measured spectra. The phase $\phi_i$ of the base function is an equally distributed MC variable between 0 and $2\pi$. In addition, $f_0(\lambda) = 1$ is used to account for constant offset. The weights $\gamma_i$ for the base functions are selected in an $N$-dimensional spherical coordinate system (Hicks and Wheeling (1959)) in such a way that the variance of the final deviation function always equals to unity. In practice, this means that the weights $\gamma_i$ are generated by scaling the random variables $Y_i \sim \mathcal{N}(0,1)$ as

$$\gamma_i = \frac{Y_i}{\sqrt{Y_0^2 + Y_1^2 + ... + Y_N^2}}. \tag{12}$$

The complete $N$-dimensional system requires orthogonal base functions, such as full periods of sine functions, to allow an arbitrary shape of deviation function $\delta(\lambda)$ with unity variance. It is possible to use other orthogonal sets of functions, such as Chebyshev polynomials instead of sinusoidal base functions, but that would involve more complicated mathematics. This is discussed in more detail in Appendix B.

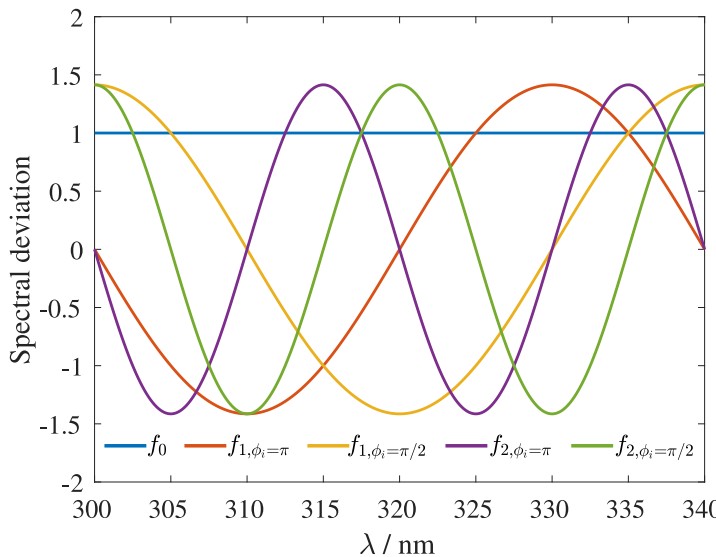

**Figure 4.** First three base functions with unity variance, $f_1$ and $f_2$ plotted with the phases $\phi_i = \pi$ and $\phi_i = \pi/2$.

The deviation functions obtained with the cumulative Fourier series are used to distort the measured spectra $E(\lambda)$ as

$$E_e(\lambda) = [1 + \delta(\lambda)\,u(\lambda)]\,E(\lambda), \tag{13}$$

where $u(\lambda)$ is the relative standard uncertainty of the spectrum. The corresponding spectral deviation is applied separately to the factors of Eq. (4), i.e., the extra-terrestrial solar spectrum, ozone absorption cross-section, and Rayleigh scattering.

Variances of the TOC values obtained by varying the weights $\gamma_i$ and the phase terms $\phi_i$ give the desired uncertainties. Figure 5(a) presents how the uncertainty induced by deviation in spectral irradiance $E(\lambda)$ (circles) changes with increasing $N$. Each standard uncertainty of TOC in Fig. 5(a) was estimated from the MC results obtained by running the TOC retrieval 1000 times so that the phases $\phi_i$ and the weights $\gamma_i$ of the base functions were independent at each round. Retrieved TOC deviations resemble the Gaussian distribution when the order of complexity of the deviation function is $N \geq 2$ as illustrated in Figs. 5(b) and 5(c). As we can see, *full* correlation with the base function $f_0(\lambda)$ at $N = 0$ causes a negligible uncertainty to TOC. The maximum at $N = 1$ gives uncertainty for an *unfavourable* case of correlations with base functions $f_0(\lambda)$ and $f_1(\lambda)$. Cases $N = 80$ for QASUME, $N = 100$ for BTS, or $N = 125$ for AVODOR correspond to the Nyquist criterion ($N = n/2$) with base functions and give the uncertainty with no spectral correlations (only *random* noise). The obtained TOC value is affected most by spectral distortion that mimics the spectral shape of the ozone absorption. The first combination of constant offset and one sinusoidal function with two sign changes within the region of interest is closest to this extreme.

The ozone absorption cross-section $\alpha_{O_3}(\lambda, T_{eff})$ is a direct multiplier of TOC, and thus the uncertainty in TOC is directly proportional to the deviations in the ozone absorption cross-section. The uncertainty in TOC arising from the spectral deviation in $\alpha_{O_3}(\lambda, T_{eff})$ is plotted as crosses in Fig. 5(a) as a function of the increasing $N$. The ozone absorption cross-section, Serdyuchenko–Gorshelev data set (Weber *et al.* (2016)), has a wavelength step size of 0.01 nm, and thus the standard

uncertainty of 0.05% at the Nyquist criterion $N = 2500$ is outside the range displayed in Fig. 5(a). Unlike the negligible effect of *full* spectral correlation in the spectral irradiance $E(\lambda)$ in TOC, *full* correlation ($N = 0$) in the ozone absorption cross-section produces approximately the same uncertainty as *unfavourable* correlation ($N = 1$). Generally, these results cannot be known before the analysis is carried out, using a method that does not have any internal limitation to the shape of the deviation

5 function $\delta(\lambda)$. In some other cases, the uncertainty extreme appears at higher $N$-values, e.g., $N = 3$ noted for correlated colour temperature by Kärhä *et al.* (2017b).

One major problem in uncertainty estimation is that typically many of the correlations in spectral irradiance data are unknown. Figure 5(a) can be used to find limits for the uncertainties assuming different correlation scenarios. In the analysis carried out in this paper, we estimate for each uncertainty component which kind of correlations it most likely has. For this, we

10 divide the correlation into three categories, *full*, *unfavourable*, and *random* and estimate fractions on the assortment of these correlations. *Full* indicates that relative deviation is wavelength independent, such as with distance setting in spectral irradiance measurements. *Random* indicates no correlation between spectral values. As can be seen in Fig. 5(a), the uncertainty caused by noise depends on the Nyquist criterion, increasing with the smaller number of base functions. *Unfavourable* indicates an unknown deviation with systematic spectral structure that produces a large deviation in TOC.

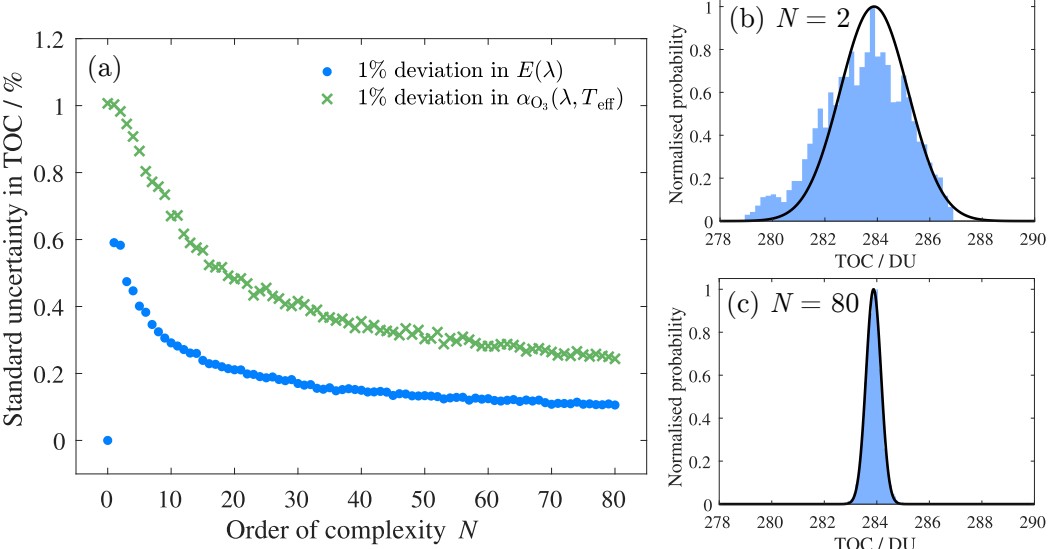

**Figure 5.** (a) Standard uncertainties of TOC at local noon as a function of the order of complexity $N$ for QASUME spectroradiometer with 1% deviation in spectral irradiance $E(\lambda)$ plotted as circles and 1% deviation in ozone absorption cross-section $\alpha_{O_3}(\lambda, T_{\text{eff}})$ plotted as crosses. The distributions of TOC values arising from 1% deviation in $E(\lambda)$ with the order of complexity of $N = 2$ in (b) and $N = 80$ in (c). The black solid curve denotes Gaussian distribution.

## 4.2 Uncertainty budgets for spectroradiometers

Uncertainty budgets of the direct solar spectral irradiance measurements for QASUME, BTS, and AVODOR are presented in Tables 1–3. The tables also state fractions that we estimate for the three correlation types introduced for each component. The uncertainties due to radiometric calibration include factors such as the uncertainty of the standard lamp used, and the additional uncertainty due to noise and alignment. QASUME has been validated using various methods, thus the uncertainty due to calibration is low, 0.55% (Hülsen *et al.* (2016)). For QASUME and BTS, we assume the correlations to be equally distributed between *full* correlation, *unfavourable* correlation, and *random* correlation (Kärhä *et al.* (2018)). Spectra measured with AVODOR are significantly noisier, thus half of the calibration uncertainty is associated to the random component. Values for instability of the calibration lamp are based on long-term monitoring. The lamp irradiances have been noted to gradually drop with no significant wavelength structure within the wavelength region concerned. Non-linearity values are estimations of the operators of the devices. Non-linearity is typically manifested so that the responsivity of the device changes gradually from high readings to low readings. This can cause significant change in the TOC values, thus we assume the correlation type to be *unfavourable*. Uncertainties due to device stability and temperature dependence are based on long-term monitoring. The changes have been found to be independent on wavelength in the region concerned, thus *full* correlation is assumed. Noise is the average standard deviation of typical measurements at noon over the wavelength region concerned. The wavelength scales of the devices have been checked using emission lines of gas discharge lamps. The uncertainty values given are the estimated standard deviations of the possible remaining errors after corrections. Wavelength error can introduce a significant change in TOC, because it introduces an error in the form of the derivative of the spectral irradiance. Thus, *unfavourable* correlation is assumed. Most of the uncertainty components are slightly wavelength dependent but to simplify simulations, average uncertainty values are used over the wavelength range between 300 nm and 340 nm.

**Table 1.** Uncertainties of the measurement for QASUME spectroradiometer.

| QASUME Source of uncertainty in measured $E(\lambda)$ | Standard uncertainty % | Correlation *full* | *unfavourable* Fraction | *random* |
|---|---|---|---|---|
| Radiometric calibration | 0.55 | $1/\sqrt{3}$ | $1/\sqrt{3}$ | $1/\sqrt{3}$ |
| 250 W lamp stability | 0.14 | 1 | 0 | 0 |
| Non-linearity | 0.25 | 0 | 1 | 0 |
| Stability | 0.60 | 1 | 0 | 0 |
| Temperature dependence | 0.20 | 1 | 0 | 0 |
| Measurement noise | 0.20 | 0 | 0 | 1 |
| Wavelength shift | 0.10 | 0 | 1 | 0 |
| Combined uncertainty ($k=1$) | 0.91% | 0.72% | 0.42% | 0.38% |

**Table 2.** Uncertainties of the measurement for BTS spectroradiometer (Zuber *et al.* (2017b)).

| BTS<br>Source of uncertainty<br>in measured $E(\lambda)$ | Standard<br>uncertainty<br>% | *full* | Correlation<br>*unfavourable*<br>Fraction | *random* |
|---|---|---|---|---|
| Radiometric calibration | 0.80 | $1/\sqrt{3}$ | $1/\sqrt{3}$ | $1/\sqrt{3}$ |
| 250 W lamp stability | 0.20 | 1 | 0 | 0 |
| Non-linearity | 0.40 | 0 | 1 | 0 |
| Stability | 0.80 | 1 | 0 | 0 |
| Temperature dependence | 0.10 | 1 | 0 | 0 |
| Measurement noise | 0.20 | 0 | 0 | 1 |
| Wavelength shift | 0.10 | 0 | 1 | 0 |
| Combined uncertainty ($k = 1$) | 1.24% | 0.95% | 0.62% | 0.50% |

**Table 3.** Uncertainties of the measurement for AVODOR spectroradiometer.

| AVODOR<br>Source of uncertainty<br>in measured $E(\lambda)$ | Standard<br>uncertainty<br>% | *full* | Correlation<br>*unfavourable*<br>Fraction | *random* |
|---|---|---|---|---|
| Radiometric calibration | 2.50 | 1/2 | 1/2 | $1/\sqrt{2}$ |
| 250 W lamp stability | 0.14 | 1 | 0 | 0 |
| Non-linearity | 0.50 | 0 | 1 | 0 |
| Stability | 0.60 | 1 | 0 | 0 |
| Temperature dependence | 0.20 | 1 | 0 | 0 |
| Measurement noise | 1.30 | 0 | 0 | 1 |
| Wavelength shift | 0.10 | 0 | 1 | 0 |
| Combined uncertainty ($k = 1$) | 2.94% | 1.41% | 1.35% | 2.19% |

## 4.3 Uncertainty budget for total ozone column

Table 4 presents an uncertainty budget for TOC that would be obtained with the high-accuracy QASUME spectroradiometer at local noon. All major uncertainty components are listed and estimated. The uncertainty components divided to the three correlation types have been analysed with the new model. The other components (a)–(d) in Table 4 have been solved using traditional MC modelling because the mechanism for the uncertainty propagating to TOC is known.

**Table 4.** An example uncertainty budget for QASUME spectroradiometer measured at local noon on the clear day of 17 September 2016. The last column states the standard deviations in $u(\text{TOC})$ corresponding to each individual uncertainty component for TOC = 284 DU retrieved from the QASUME spectrum using the spectral range of 300 nm – 340 nm at the solar zenith angle of 26.35°. The expanded uncertainty stated, $U(\text{TOC}) =$ 3.6 DU, has been obtained by multiplying the combined standard uncertainty with a coverage factor $k = 2$.

| Source of uncertainty | Standard uncertainty in $E(\lambda)$ | in exponent | Correlation *full* | *unfavourable* | *random* | $u(\text{TOC})$ |
|---|---|---|---|---|---|---|
| | % | % | | Fraction | | DU |
| **Measurement** | | | | | | |
| Radiometric calibration | 0.55 | | $1/\sqrt{3}$ | $1/\sqrt{3}$ | $1/\sqrt{3}$ | 0.44 |
| 250 W lamp stability (one year) | 0.14 | | 1 | 0 | 0 | 0.00 |
| Non-linearity | 0.25 | | 0 | 1 | 0 | 0.33 |
| Stability | 0.60 | | 1 | 0 | 0 | 0.00 |
| Temperature dependence | 0.20 | | 1 | 0 | 0 | 0.00 |
| Measurement noise | 0.20 | | 0 | 0 | 1 | 0.06 |
| Wavelength shift | 0.10 | | 0 | 1 | 0 | 0.13 |
| **Uncertainties related to $E(\lambda)$** | | | | | | |
| Extra-terrestrial spectrum (Gröbner *et al.* (2017)) | 1.00 | | $1/\sqrt{3}$ | $1/\sqrt{3}$ | $1/\sqrt{3}$ | 0.95 |
| **Uncertainties related to exponent of Eq. (4)** | | | | | | |
| O$_3$ cross-section (Weber *et al.* (2016)) | | 1.5 | 0.23 | 0.23 | 0.95 | 1.41 |
| Rayleigh scattering (Bodhaine *et al.* (1999)) | | 0.1 | $1/\sqrt{3}$ | $1/\sqrt{3}$ | $1/\sqrt{3}$ | 0.09 |
| O$_3$ layer altitude of 26 km, $u = 0.5$ km | | (a) | | | | 0.01 |
| Rayleigh layer altitude of 5 km, $u = 0.5$ km | | (b) | | | | 0.00 |
| Temperature of O$_3$ cross-section at 228 K, $u = 1$ K | | (c) | | | | 0.28 |
| Station pressure of 772.8 hPa, $u = 1.3$ hPa | | (d) | | | | 0.05 |
| | | | | | $U(\text{TOC})$ | **3.6** |

(a) Air mass $m_{\text{TOC}}$ varies as a function of the altitude of O$_3$ layer.

(b) Air mass $m_{\text{R}}$ varies as a function of the altitude of Rayleigh scattering layer.

(c) O$_3$ cross-section varies as a function of temperature.

(d) Rayleigh scattering depends on the station pressure.

The uncertainties produced in TOC were obtained separately for all components, by setting other uncertainties to zero. Division of the correlation to the three categories introduced are stated for each row as fractions $r_{full}$, $r_{unfav}$, and $r_{random}$. For

example, the ground-based spectrum $E(\lambda)$ measured is deviated with the three correlation components as

$$E_{e}(\lambda) = (1 + u \, r_{full} f_0(\lambda)) \cdot \left(1 + u \, r_{unfav} \sum_{i=0}^{1} \gamma_i' f_i(\lambda)\right) \cdot \left(1 + u \, r_{random} \sum_{j=0}^{N} \gamma_j'' f_j(\lambda)\right) E(\lambda), \qquad (14)$$

where $\gamma_i'$ and $\gamma_j''$ are independent MC variables generated using Eq. (12).

It is worth noting that not all uncertainty components affect the spectrum $E(\lambda)$ directly, but via the exponent of Eq. (4).

Corresponding formulas are used to evaluate the effect of uncertainties in extra-terrestrial solar spectrum, ozone absorption cross-section, and Rayleigh scattering. The last column in Table 4 states the standard uncertainties produced by each uncertainty component with the assumed fractions, calculated with an irradiance spectrum measured at local noon with QASUME (Hülsen *et al.* (2016)). The expanded uncertainty of the TOC, obtained as the square sum of the individual components and multiplied with coverage factor $k = 2$, for this spectral measurement is 3.6 DU (1.3%).

The QASUME spectroradiometer has a combined measurement standard uncertainty of 0.91% (Hülsen *et al.* (2016)) arising from the uncertainty components explained in Section 4.2. The uncertainty components stated are typical in solar UV spectral irradiance measurements (Bernhard and Seckmeyer (1999)). Division of the radiometric calibration uncertainty to equal fractions of $1/\sqrt{3}$ is based on typical spectral correlations noted in intercomparisons (Kärhä *et al.* (2018)). The lamp data obtained from national standard laboratories are highly accurate but also typically spectrally correlated. Due to very low noise,

elements such as interpolation functions, offsets and slopes are present in the data. When the calibration is transferred further, uncertainty increases due to noise, and correlations reduce. We thus assume that in this high accuracy calibration, there are equal amounts of *fully* correlated, *unfavourably* correlated, and *uncorrelated* uncertainties included.

For $E_{ext}(\lambda)$, we use QASUME-FTS (Gröbner *et al.* (2017)). We assume the correlation to be similar to a standard lamp, thus containing equal fractions of *full*, *unfavourable*, and *random* correlations. The QASUME-FTS is provided in air wavelengths

with a step size of 0.01 nm. Otherwise, the wavelength shift due to vacuum-air interface should be corrected from the extra-terrestrial spectrum.

As the reference ozone absorption cross-section, the Serdyuchenko–Gorshelev data set given in air wavelengths with a step size of 0.01 nm, was used with 1.5% standard uncertainty (Weber *et al.* (2016)). The systematic and random uncertainties of the Serdyuchenko–Gorshelev data set are given separately (Weber *et al.* (2016)). We further estimate that the systematic

uncertainty given may include equal fractions of *fully* correlated and *unfavourably* correlated uncertainty. Thus, according to that we use the fractions of 0.23 for *full*, 0.23 for *unfavourable* and 0.95 for *random* correlations. *Full* correlation with a fraction of 0.23 produces a standard uncertainty of 0.96 DU, *unfavourable* correlation with a fraction of 0.23 produces a standard uncertainty of 1.01 DU and *random* correlation with a fraction of 0.95 produces a standard uncertainty of 0.22 DU. Altogether, the ozone cross-section causes an uncertainty of 1.41 DU to TOC, and is thus the dominating component in the

uncertainty. If the fractions of correlations are not equally distributed between *full* and *unfavourable*, uncertainty in TOC does not change significantly from 1.41 DU. Fractions of 0.31 for *full*, 0 for *unfavourable* (or fractions of 0 for *full*, 0.31 for *unfavourable*), and 0.95 for *random* correlations, cause an uncertainty of 1.33 DU (or 1.43 DU) in TOC.

For components (a)–(d) in Table 4, the mechanism of contributing to the uncertainty of TOC is known. We know the standard uncertainty of the $O_3$ layer altitude of 26 km to be $u = 0.5$ km, so we vary the altitude accordingly and note the variance of the resulting TOC. Rayleigh scattering and aerosols are set at the altitude of 5 km $\pm 0.5$ km, which influences the relative air mass $m_R \approx m_{AOD}$ (Gröbner *et al.* (2017)). This component causes negligible uncertainty to TOC. For calculating $\tau_R(\lambda, P_0, z_0, \phi)$, we use a model by (Bodhaine *et al.* (1999)) with 0.1% uncertainty with equal estimated fractions of correlation types. The correlation has been obtained by studying how this data deviates from the model by (Nicolet (1984)). The ozone and temperature profiles were measured with a sonde during the campaign and based on the profiles the effective altitude of the ozone layer was at 26 km $\pm$ 0.5 km at the effective temperature of 228 K $\pm$ 1 K. The effect on TOC was obtained by randomly varying the altitude with the Gaussian uncertainty distribution. The same applies to air pressure that was 772.8 hPa with a standard uncertainty of 1.3 hPa. The effect of temperature on TOC was obtained by randomly varying the temperature with its standard uncertainty of 1 K. Varying the temperature systematically changes the spectral ozone absorption cross-section according to Eq. (6).

Stray light that affects TOC results at large solar zenith angles (Appendix A) has not been accounted for in the uncertainty analysis due to lack of information. Proper correction and estimation of the uncertainty due to stray light would require the stray light correction matrix to be measured. The effect of stray light, typical for measurements with array spectroradiometers, was reduced from TOC results by fitting BTS and AVODOR spectra with the OLS method. Then, these TOC values were corrected to be compatible with the RLS results by Eq. (9). The correction factor involves standard uncertainty that is 0.1% (0.28 DU) for BTS and 1.1% (3.10 DU) for AVODOR.

## 5   Results and discussion

The calculated TOC values obtained by the three different spectroradiometers on 17 September 2016 are presented in Fig. 6. Expanded uncertainties of the TOC values calculated are stated in DU as error bars. Measurement results of Brewer spectrophotometer #183 used as a reference in the intercomparison have been included in Fig. 6 as well. Looking at the absolute values of TOC in Fig. 6, we may conclude that the results of QASUME and Brewer #183 are in excellent agreement. Also the TOC values estimated for BTS and AVODOR are in agreement with Brewer #183 within uncertainties.

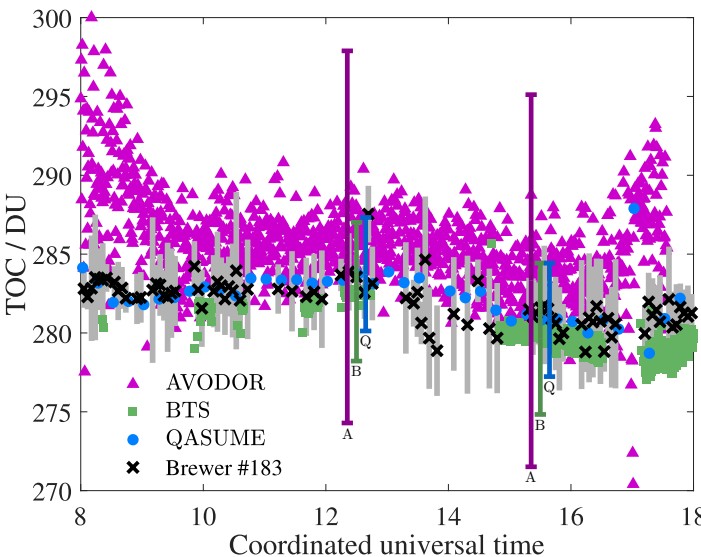

**Figure 6.** Total ozone columns (TOC) derived from the solar UV spectra from 300 nm to 340 nm with expanded uncertainty bars ($k = 2$) for QASUME indicated as blue circles, BTS indicated as green squares, and AVODOR indicated as magenta triangles. The TOC measured with the Brewer #183 is plotted as black crosses with grey uncertainty bars ($k = 2$).

The relative uncertainties of the TOC values obtained with the three instruments are shown in Fig. 7. The expanded uncertainties of the TOC data sets at local noon are 3.6 DU (1.3%) for the QASUME spectroradiometer, 4.4 DU (1.5%) for the BTS spectroradiometer, and 13.3 DU (4.7%) for the AVODOR spectroradiometer.

It is of interest to compare the obtained uncertainties with values assuming no correlations. If we neglect correlations, i.e.,

we assume the fractions in Table 4 to be 0 for the *full* and *unfavourable* correlations and 1 for the *random* correlation, and run the simulations with the spectrum measured at local noon, we obtain the expanded uncertainty $U_{\text{QASUME}}(\text{TOC}) = 0.9$ DU (0.3%), $U_{\text{BTS}}(\text{TOC}) = 1.1$ DU (0.4%), and $U_{\text{AVODOR}}(\text{TOC}) = 7.7$ DU (2.7%). These values are on average a factor of 3 lower than the uncertainties accounting for correlations. This analysis assumes random noise only.

A typical practice in an analysis like this is to add a component introduced by the standard deviation of the fit to the

uncertainty. The standard uncertainty to be added to $u(\text{TOC})$ because of the standard deviation of the fit is 0.2 DU with QASUME, 0.7 DU with BTS and 3.4 DU with AVODOR, raising the corresponding total expanded uncertainties to 1.0 DU (0.3%), 1.8 DU (0.6%), and 10.3 DU (3.6%). The results are generally in agreement within these lower uncertainties as well. However, comparing differences in the TOC results of the different spectroradiometers does not represent the uncertainty in the absolute TOC scale, since the ozone retrieval algorithm uses the same extra-terrestrial spectrum and ozone absorption

cross-section for all the instruments. Changing the extra-terrestrial spectrum or the ozone absorption cross-section to another data set may shift all the TOC values of the instruments beyond the latter low uncertainties.

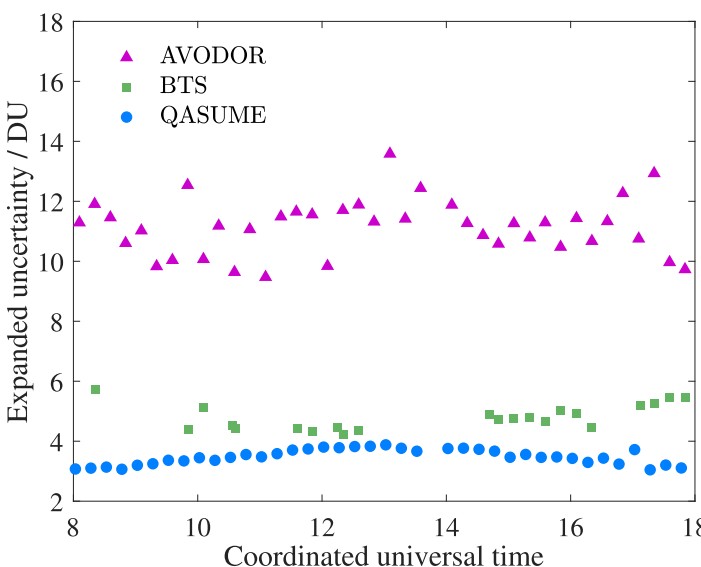

**Figure 7.** Relative expanded uncertainties of the total ozone columns derived from QASUME, BTS, and AVODOR spectra obtained by the relative least squares fitting method.

Figure 6 shows that the effect of stray light can be effectively reduced from TOC results by using the OLS fitting described in Appendix A with the correction in Eq. (9). For example, BTS and QASUME results are in good agreement even at the largest zenith angles. TOC results of AVODOR are in agreement at noon, but the results before 09:00 and after 17:00 deviate from the other instruments by 10 DU.

## 6 Conclusions

In this work, we introduced one possible way to take into account spectral correlations in the uncertainties of the atmospheric ozone retrieval and estimated the TOC uncertainties obtained from the spectral data of three different spectroradiometers, measured at the ATMOZ field measurement campaign at the Izaña Atmospheric Observatory. It should be noted that the method proposed has a drawback that the unknown correlations have to be approximated based on experience. However, the method has merits in estimating the order of magnitude of possible uncertainties accounting for correlations. The typical assumption made, that uncertainties are spectrally uncorrelated, is just an assumption as well, and in many cases not valid. The uncertainty values obtained with the new model are higher than the uncertainties obtained with the traditional method neglecting correlations because some of the major uncertainty components may contain systematic spectral deviations. These results demonstrate the importance of accounting for correlations. If their origins and magnitudes are known, they can be accounted for precisely using methods of (JCGM 100 (2008)).

The new model uses similar approach to our previously developed MC uncertainty model for correlated colour temperature (CCT) (Kärhä *et al.* (2017b)). In the article, we demonstrated the use of the model for calculating the CCT of a Standard

Illuminant A. For Standard Illuminant A, the case representing uncertainty with *unfavourable* correlations in CCT was found at $N = 3$. On the contrary, for the ozone retrieval the deviation at $N = 1$ produces the largest uncertainty, which is in a way trivial compared with CCT. The use of a set of sine functions as base functions was originally developed for the more complicated situation of CCT where it was not known where the *unfavourable* uncertainty would be. When we now have analysed the situation, an uncertainty arising from unfavourable correlations in the ozone retrieval could as well be modelled e.g. using a combination of *full* spectral deviation, a simple slope, and a parabola as the deviation function mimicking *unfavourable* correlations. This is discussed in more detail in Appendix B.

The new MC method for estimating uncertainties in TOC in the presence of systematic spectral deviations provides more complete estimations of the uncertainty budget compared with the traditionally used methods. The TOC values retrieved from different instrument data were well in agreement within the uncertainties estimated with the new MC method. Although the TOC results obtained using different instruments have good agreement, these differences do not represent the uncertainty of the absolute TOC scale. The TOC uncertainties we have estimated cover also possible offsets in the absolute TOC scale, arising from the uncertainties in the ozone absorption cross-section and extra-terrestrial spectrum that are the dominating components in the uncertainty budget.

## Appendix A: Selecting the least squares fitting method

Two of the instruments being compared suffer from stray light and noise that distort the TOC results. In earlier studies e.g. by Herman *et al.* (2015), stray light in array spectroradiometers has been noted to decrease TOC values at large solar zenith angles resulting in an inverted U-shape dependence of the diurnal TOC variation. The effect of stray light can be compensated by reducing the effect of short-wavelength tail, either by limiting the wavelength range or the dynamic range used, or by weighting the results. We studied various weightings and selection methods for data in order to find an objective way to perform the ozone retrieval method and to analyse the results.

Figure A1 shows the TOC results as a function of time analysed with three different weighting methods for least squares minimisation. The methods include a relative least squares fitting (RLS) in Eq. (8) with $w(\lambda) = E(\lambda)^{-2}$, RLS fitting with the dynamic range (DR) limited to avoid stray light, and an ordinary least squares fitting method (OLS) with $w(\lambda) = 1$.

TOC values estimated for QASUME in Fig. A1(a) have no significant solar zenith angle dependence, and limiting the dynamic range of RLS fitting only affects individual TOC values in the early morning and late afternoon compared with RLS fitting over the complete spectral range. Using OLS fitting method, the diurnal variation of the TOC remains, but the values are underestimated by a constant factor of 1.013.

TOC values estimated for BTS in Fig. A1(b) and AVODOR in Fig. A1(c) have severe dependence on the solar zenith angle when using RLS fitting method. The solar zenith angle dependence decreases to approximately half when limiting the dynamic range to exclude the baseline noise from the fitting. Best results are obtained by using the OLS minimisation which practically removes the solar zenith angle dependence for both BTS and AVODOR but introduces an offset to TOC results. Almost similar offset in TOC results was noted for QASUME. The OLS method violates against the heteroskedasticity of our data sets, since

we know that the absolute spectral uncertainties are not constant at the wavelength range studied. The only reason to use the OLS method for array spectroradiometers is to reduce the effect of stray light from TOC results. Hence, we correct the TOC results obtained with the OLS method with correction factors estimated from the ratios of TOC values determined from the local noon spectra using both RLS and OLS methods. The correction factors were averaged over 10 samples around noon, being 1.006 for BTS and 1.013 for AVODOR with standard deviations of 0.1% (0.28 DU) and 1.1% (3.10 DU), respectively. This correction makes the TOC results comparable with devices analysed using the RLS method.

For QASUME, we use RLS minimisation with the dynamic range limited to four orders of magnitude, as it uses most of the useful undistorted data. This is also consistent with the earlier methods used with monochromator-based spectroradiometers e.g. by Huber *et al.* (1995).

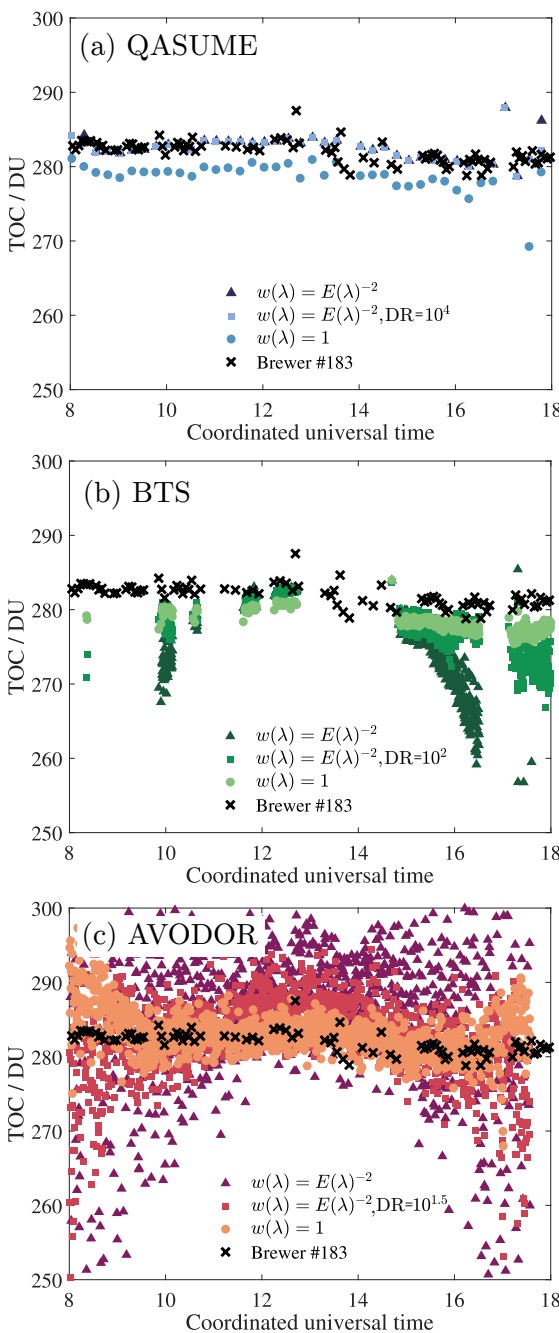

**Figure A1.** TOC values during the day estimated using different weightings in the least squares minimisation for QASUME (a), BTS (b) and AVODOR (c). TOC values for Brewer #183 are plotted as black crosses for comparison. The colour codes and the associated figure legends denote the weighting and the dynamic range (DR) used.

## Appendix B: TOC uncertainties obtained using Chebyshev polynomials

In the MC uncertainty analysis, it is possible to use other orthogonal sets of functions instead of sinusoidal functions, such as Chebyshev polynomials shown in Fig. B1. A Chebyshev polynomial of the first kind $T_j(\lambda)$ of order $j$ (Kreyszig (2006), p. 209) is defined as

$$T_j(\lambda) = \cos\left[j\arccos\left(\frac{2\lambda - \lambda_1 - \lambda_2}{\lambda_2 - \lambda_1}\right)\right], \tag{B1}$$

where $\lambda_1$ is the short wavelength limit and $\lambda_2$ is the long wavelength limit for the spectra measured. To create arbitrary spectral deviations with unity variance, each Chebyshev polynomial, except for $g_0(\lambda) = T_0(\lambda) = 1$, needs to be normalised to unity variance as

$$g_j(\lambda) = \frac{T_j(\lambda)}{\sigma_j}, \tag{B2}$$

where $\sigma_j$ is the standard deviation of $T_j(\lambda)$. In practice, Chebyshev polynomials in Fig. (B1) can be generated fast using recurrence (e.g. Fateman (1989)) as

$$T_j(\lambda) = 2\left(\frac{2\lambda - \lambda_1 - \lambda_2}{\lambda_2 - \lambda_1}\right)T_{j-1}(\lambda) - T_{j-2}(\lambda), \tag{B3}$$

where $T_0(\lambda) = 1$, and $T_1(\lambda) = (2\lambda - \lambda_1 - \lambda_2)/(\lambda_2 - \lambda_1)$ is a straight line. The scaling by the standard deviation according to Eq. (B2) is performed after generating the Chebyshev polynomials.

Each base function of the cumulative Fourier series in Eq. (10) was formed with sine (odd) and cosine (even) terms (Kreyszig (2006), p. 628),

$$\sin\left[i\left(2\pi\frac{\lambda - \lambda_1}{\lambda_2 - \lambda_1}\right) + \phi_i\right] = \cos(\phi_i)\cdot\sin\left(2\pi i\frac{\lambda - \lambda_1}{\lambda_2 - \lambda_1}\right) + \sin(\phi_i)\cdot\cos\left(2\pi i\frac{\lambda - \lambda_1}{\lambda_2 - \lambda_1}\right), \tag{B4}$$

where the phase $\phi_i$ is an equally distributed MC variable between 0 and $2\pi$. Hence, the analysis based on Chebyshev polynomials to be compatible with the sinusoidal approach, each base function $f_i(\lambda)$ with index $i = (j+1)/2$ is formed by the combination of odd ($j = 1, 3, 5, ..., n-1$) and even ($j + 1 = 2, 4, 6, ..., n$) terms as

$$f_i(\lambda) = \cos(\phi_i)\cdot g_j(\lambda) + \sin(\phi_i)\cdot g_{j+1}(\lambda), \tag{B5}$$

where the weights $\cos(\phi_i)$ and $\sin(\phi_i)$ set the variance of $f_i(\lambda)$ to unity. These base functions $f_i(\lambda)$ can be used with Eqs. (10) and (13).

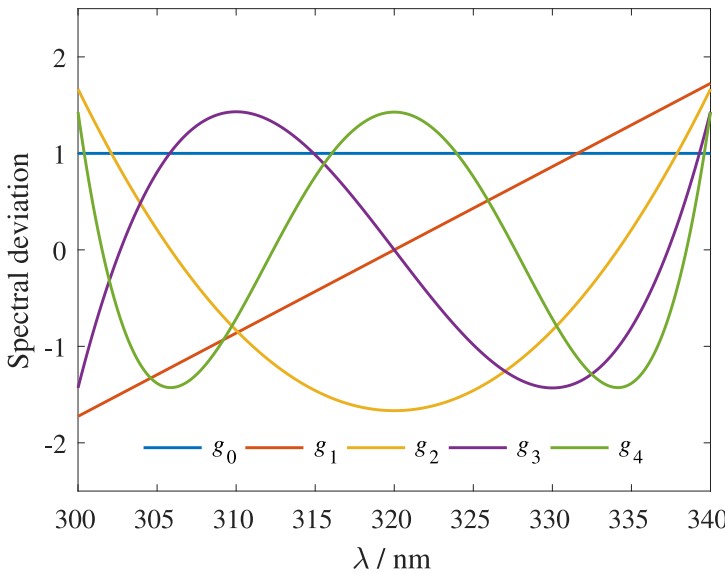

**Figure B1.** First five Chebyshev polynomials with unity variance corresponding to Fig. 4 with sinusoidal base functions.

Figure B2 compares standard uncertainties of TOC obtained by generating arbitrary spectral deviations using Chebyshev polynomials as base functions (circles and triangles), formed using Eq. (B5), with those obtained by using sinusoidal base functions (crosses). The uncertainties change slightly at the lowest complexity orders of deviations when sinusoidal base functions are replaced with Chebyshev polynomials, but essentially the results are similar.

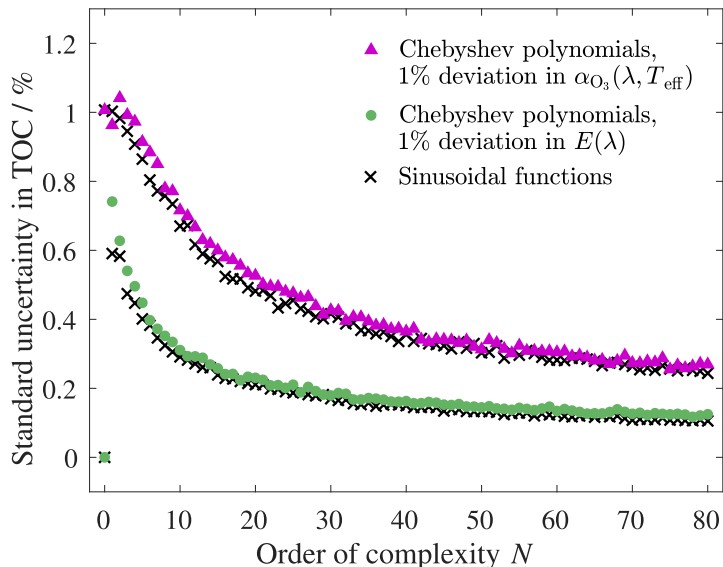

**Figure B2.** Standard uncertainties of TOC simulated using a local noon spectrum of QASUME with Chebyshev polynomials as base functions. The TOC uncertainties with the input standard deviation of 1% in the spectral irradiance values are shown as green circles and the TOC uncertainties with the input standard deviation of 1% in the ozone absorption cross-section are shown as magenta triangles. Standard uncertainties simulated with sinusoidal base functions (black crosses) from Fig. 5(a) are plotted for comparison.

*Acknowledgements.* Peter Sperfeld from PTB is acknowledged for measuring and providing the BTS data set. Alberto Redondas and all personnel from Izaña Atmospheric Research Center, AEMET, Tenerife, Canary Islands, Spain, are acknowledged for measuring and providing the Brewer #183 data set and the environmental parameters such as the sonde data. A.V. is grateful for the incentive grant by the Emil Aaltonen Foundation, Finland. This work has been supported by the European Metrology Research Programme (EMRP) within the joint research project ENV59 "Traceability for atmospheric total column ozone" (ATMOZ). The EMRP is jointly funded by the EMRP participating countries within EURAMET and the European Union.

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
