# Peer review of "Uncertainty analysis of total ozone derived from direct solar irradiance spectra in the presence of unknown spectral deviations"

_Atmospheric Measurement Techniques, 2017_

## Referee Comment (RC1) · Anonymous Referee #4 · 20 Feb 2018

The article by Vaskuri et al presents a novel method for the determination of the uncertainty of total ozone measurements. Although the findings of the study are interesting, there are parts of the manuscript that have to be improved prior to its publication. In the following there is a list of more specific comments.

Abstract

P1,L1: replace "calculate" with "estimate".

Introduction

P1, L13-14: At this point the authors should make clear that they are talking for correlations in spectral measurements. According to the authors this is the main problem solved when the new methodology is applied. I also suggest adding more information here to help the reader understand what they mean when they refer to "correlations".

P2, L4-5: Do you mean here that the field of view of the spectroradiometers is equal to exactly one solar diameter? If not, then some scattered irradiance also enters the spectrometer.

Section 2

The tables 1, 2 and 3 are presented here without any discussion regarding the presented quantities. I suggest that they should be moved to the uncertainty estimation section (section 4). Furthermore, some discussion (e.g. explaining the presented correlation types, description of how the different uncertainty types were estimated) would be useful.

Section 3

P7, L2: Gröbner and Kerr (2001) did not assume that the air mass factors for aerosols and Rayleigh scattering are equal.

Section 4

P14, L5: Again, the reference of Gröbner and Kerr (2001) is not correct here.

Section 5

P17: Please add more information regarding the linear model used for AOD. E.g., why using the particular model for AOD? Are a and b the same with those of Ångström (1964)? If not, how they are estimated? What happens if the TOC is derived by QASUME and BTS using this linear AOD model?

Conclusions

P18, L27: The results from AVODOR deviate up to 10 DU (and not 10 DU) depending on the SZA. Is the stray light effect enough to explain these discrepancies?

The last paragraph of the conclusions section is now written leads to the conclusion that the main outcome of the study is that the AVODOR is not suitable for TOC measurements, while QASUME and BTS are. In my opinion the main outcome of this study is that the presented method provides more accurate estimations of the uncertainty budget compared to the traditionally used methods. However, it is not adequate for properly estimating uncertainties if the instruments are not characterized for systematic measurement errors. I suggest re-writing the conclusions section in a way that the main conclusions of the study are highlighted.

---

## Referee Comment (RC2) · Anonymous Referee #3 · 23 Feb 2018

The article by Vaskuri et al presents a novel method for implementing correlations in measurement uncertainty calculations on the example of total ozone retrieval using spectral irradiance measurements. The approach presented is very promising and the high complexity of the scientific work dealing with correlated spectral irradiance measurements is of major importance for the community. The experimental and scientific work is on a very high level including a thorough characterization and understanding of the uncertainties of the applied instruments and methods. However, some of the scientific details are expressed incomprehensible and in some cases more details are needed prior to publication.

Specific comments:

P.4. Table1 1. – 3. The table for Uncertainties including the fraction of correlation need further explaining, especially since the issue of correlation is introduced much later in section 4. If these uncertainties have been published in that form earlier, quotation would be helpful in the figure caption. If not, these tables should be moved to section 4 prior to table 4 or in combination with table 4 since the explaining is done on p.15 L4-8. Why is uncertainty of radiometric calibration of AVODOR so much higher than for the other instruments? Just because of low SNR in the UV region? The stated uncertainties given in the tables are valid for the whole spectral range of the instrument? I would expect uncertainties related to radiometric calibration and measurement noise to be wavelength dependent. Or are these stated values the upper limit of uncertainties? In the following text, there is a lot discussion about straylight effects. However, there is no explicit uncertainty component related to straylight or straylight correction?

P. 7, L15 The least square fitting might lead to local minima instead of the global minimum. How is that accounted for? Especially if AOD and TOC are both fitting parameters.

P.9, L20 "In this paper, we do this for all components, where the mechanism of contributing to the uncertainty of TOC is known." I guess these components are those with correlation "full" and "random"? This could be specified here.

P.11, Figure 4 That figure is a bit confusing. For underlining the statement on full, unfavorable and random correlation the display of one colored graph is sufficient. The additional information gained from the u=5% and u=2.5% graph, as well as the black solid lines is not explained in the plain text and incomprehensible explained in the figure caption.

---

## Referee Comment (RC3) · Anonymous Referee #2 · 3 Mar 2018

**GENERAL COMMENTS**

The paper by Anna Vaskuri aims at calculating the uncertainty of ozone retrievals from measurements of direct irradiance spectra taking into account correlations between spectral irradiance data. The unknown total ozone content is assumed to be retrieved by relative least squares fitting of the measuremed spectrum to a modeled spectrum (Eq. 8) and possible, spectrally-correlated systematic deviations are reproduced using three terms (full, unfavourable and random) of a cumulative Fourier series. This uncertainty model is applied to calculate the contribution of some components of the total uncertainty. The deviations of the retrieved ozone from the test instruments to

the ozone retrieved by a reference MkIII Brewer are briefly reported at the end of the manuscript.

In my opinion, the paper cannot be accepted in its current form and major revisions are required.

First, the paper was probably written very quickly and important information is missing, which makes the understanding quite difficult for the unexperienced reader:

- a complete introduction about the importance of ozone measurements and the assessment of their uncertainty should be included (it is only the first line, so far);

- it is not explained why spectral data should contain correlations (physical basis);

- the only characteristics of the instruments described in the manuscript are their spectral range. I believe that a study of the instrumental uncertainties should provide a thorough description of the instruments;

- a Montecarlo model was employed, but important details such as the number of samples that were used, the obtained statistical distribution, etc. are not mentioned;

- the uncertainty components written in the tables are not properly motivated in Sect. 5. Each number should be accompanied by a clear explanation;

- the discussion about the deviation of the three instruments is very superficial (e.g., "... the reason a the systematic deviation either in the linearity, stray light properties, or the calibration of the device") and inconclusive. How is it connected to the main topic of the paper?

I also found many inaccuracies:

- language inaccuracies are listed in the "Technical corrections" section. One for all: "Izaña results" in the title is very generic. At the Izaña atmospheric observatory (not simply "Izaña"), several activities are organised and the title should appropriately tell which campaign was taken into consideration;

- there is a persistent interchange of terms that should not be mixed: uncertainty, deviation, error, etc. (e.g., "uncertainty induced by deviation", p. 10 line 9);

- QASUME is not only a "high-quality reference instrument ... at PMOD/WRC": it is the World reference UV spectroradiometer! Anyway, it should be explained how a global irradiance instrument could measure direct solar irradiance spectra (p. 2 line 5: how was the field of view "limited"?);

- the retrieval method is NOT "consistent with the ozone measurements with Brewer" (cf. specific comment 1).

Secondly, and more seriously, I have several scientific concerns that are listed in the following section. Most of all, if a bold scientific basis of the uncertainty model is not provided (e.g. why the instrumental errors should behave like a full-period sinusoid), the manuscript only represents a theoretical exercise without any advantage for real-life users.

SPECIFIC COMMENTS

1/ Retrieval model

It should be explained why the retrieval method (Eq. 8) was chosen. An obvious drawback is that this method is not invariant for spectrally-constant factors ("full correlation" or systematic errors in the absolute calibration, as hypothesised for AVODOR). It should be stated what networks / instruments use this model. For example, it is false that "This approach is consistent with the ozone measurements with Brewer" (p. 7, line 17): the Brewer algorithm is invariant for "full spectral correlation" (therefore, the "full correlation" term would not make sense with a different retrieval method, e.g. for a method where the offset is also included in a DOAS-like fit). Also, since this method gives more weight to lower irradiances, the authors should carefully explain how the "wavelength region where the signal is above the noise floor" was determined (5E-3 and 1E-5 noise floors);

**2/ Uncertainty model**

The uncertainty model (Eq. 9-10) is very complex. However, for the most part of the paper, only three terms of the Fourier series are used. Thus, I wonder why such a complex initial framework must be described. Also, it should be explained why deviations in measurements should follow this model. Coming to the "unfavourable correlation", "The obtained TOC value is affected most by spectral distortion that mimics the spectral shape of the ozone absorption... The first combination of constant offset and one sinusoidal function ... is closest to this extreme". However, from Eq. 10, this term varies with lambda1 and lambda2, so the width of the sinusoid is different for the three instruments and comparison of the results is difficult. Also, why should this term be a full sinusoid cycle? Why should the period of the oscillation be the same for all uncertainty components (calibration, measurement, cross sections, etc.)? Finally, it is obvious that phi has an enormous role for the "unfavourable correlation" term: depending on phi's value (i.e. similarity with ozone spectral cross section), the induced error could be huge or negligible. The role of phi should be explained better. Physically, what does phi represent? It is expected to randomly change in one instrument or not?

TECHNICAL CORRECTIONS

- p. 1 line 2, "directional irradiance": do you mean "direct irradiance"?

- p. 1 line 2, "correlations in the spectral irradiance data" is too generic: do you mean correlation of data within the same spectrum and measured at different wavelengths?

- p. 1 line 20, "analyse uncertainties": uncertainty of what? I guess in ozone retrievals...;

- p. 2 line 1-2: is the order in which the instruments are described (QASUME, AVODOR and BTS) the same as in the abstract (high-end scanning spectroradiometer, high-end array spectroradiometer and roughly adopted instrument)? If not, please avoid confusing the reader;

- p. 2 line 11: "total ozone content" is mentioned since the beginning of the paper, but formally defined only in page 6 (Eq. 4). Can you move Eq. 4 a bit earlier?

- p. 3 line 1, "vertical profiles were not implemented": what do you mean?

- p. 3 line 2, "shift the absolute values": values of what? "but should not have an effect": can you justify this hypothesis?

- p. 3 line 10, "the uncertainties ...are standard deviations": standard deviation of what series/samples? Can you mention which kind of measurements were employed?

- p. 3 line 11, "One of the instruments was the QASUME...": already said;

- p. 3 line 14, "every 15 minutes": explain that use of a scanning radiometer involves slower measurements, and why;

- p. 3 line 15: what is the spectral range of AVODOR? It is legitimate to say that an instrument "has been corrected"?

- p. 4 Table 1: is this table useful? The same numbers are repeated in Table 4. Also, provide bibliographic references about how each term in the "Standard uncertainty" column was calculated;

- p. 5 line 2, "fitting the ozone retrieval": can a single quantity be fitted?

- p. 5 line 5, "affects uncertainties with a factor of sqrt(N)": do you mean 1/sqrt(N)? If so, why the Brewer - which measures the irradiance for the ozone retrieval at only 4 wavelengths - is considered a reference in the paper?

- p. 6 line 1-2: give credit to Bouguer, Lambert and Beer (not Huber et al. 1995);

- p. 6 Eq. 3: a reference to the used extraterrestrial spectrum (QASUME-FTS) should be mentioned just after the equation;

- p. 6 line 12: theta is the angle at the observing site, not the angle betwen vacuum-to-air interface;

- p. 7 line 10: the extinction coefficient is defined as dTau/dz, thus it has nothing to do with beta;

- p. 7 line 11: avoid the expression "terrestrial spectrum", the radiation is from the sun, not from the Earth. Use "solar spectrum at the Earth surface";

- p. 7 line 22, "As can be seen, the signal-to-noise ratios ... differ": how can I see it from the figure, without any explanation?

- p. 9 line 3: "noise" usually defines a random variable, while stray light is a systematic effect. Don't put them together in the same sentence;

- p. 10, Eq. 11: define "u";

- p. 10 line 20, "does not have any internal limitation to the shape of the error function": what do you mean?

- p. 11 line 3, "components stating fractions": what do you mean?

- p. 12 Table 4: why is "X" used instead of "TOC"?

- p. 12 line 2: was "r" defined?

- p. 13 line 20, "a wavelength shift will introduce unfavourable correlations": why? The ozone cross section has a complex shape;

- p. 13 lines 24-25, "the wavelength shift... should be corrected for the extraterrestrial spectrum": or vice-versa?
* * *

---

## Referee Comment (RC4) · Anonymous Referee #1 · 5 Mar 2018

General Comments – This is a useful study presenting a new approach to more fully assess the uncertanties associated with measuring total ozone column with different instruments. Its advantages are that a more complete uncertainty assessment is arrived at, though the method does rely on subjective apportioning of each type of error to random, correlated or unfavourable categories. Some points of improvement would make this a very useful study for both users of ground-based data and the instrument community e.g. which are the main sources of uncertainty for each instrument type and how do these compare with the often quoted intercomparison uncertainty budgets.

Specific Comments: – P1 L14: "The reason often is that the correlations are not

known". It would be better to insert "unknown" into the previous sentence before "correlations" and remove this sentence entirely.

P1 L17: This would be better phrased as a complete uncertainty budget being necessary to understand long term environmental trends, rather than increased uncertainties improving the reliability of long term trends.

P2 L5: Better as "...excluding the remainder of the sky". Also should state field of view of each instrument here or a later point in the field campaign section.

P2 L6: This section may be better called "The Izaña field measurement campaign and instrument description", and include some additional details on each instrument - e.g. the field of view and other pertinent details. Alternatively the uncertainty tables would be better moved to later in the manuscript where the individual contributions are discussed.

P2 L15: "mountain Teide ...." » "Mount Teide at *an* altitude..."

P2 L16 "Station pressure of 772.8 hPa was monitored during the campaign with a standard uncertainty of 1.3 hPa" » "Station pressure was monitored during the campaign and determined to be 772.8 hPa with a standard uncertainty of 1.3 hPa"

P4 L4: "The tables also give division of the uncertainty components to different correlation types as described in Section 4." » "The tables also attrribute uncertainty contributions to different correlation types as described in Section 4." I think this is what you mean, either way needs a rephrasing to clarify.

P10 L2 "equal *to* unity"

P10 This section on the MC description is clearly the core of the study and where the error estimates are derived, but needs more work and clarification. The details and reasoning behind the approach may be in Karha et al 2017, but it would assist reader of this manuscript to relate MC model and, for example, its sinusoidal terms to physical sources of uncertainty, and how these are calculated for random, unfavourable, and full

correlations. At present this isn't clear.

P16 L3 If Brewer #183 is included as a reference instrument, then it would be useful to include a similar uncertainty budget for this instrument, even if only summarised. Also the community would find it useful to put these results into context and comparison with those observed at instrument intercomparisons, and often quoted as a measure of instrument or data quality. i.e. for Brewers is the actual uncertainty determined by the MC methodology much large than expected from the intercomparison error, and, what is the primary source (so efforts can be made to reduce it).

---

## Author Comment (AC1) · 22 Mar 2018

**Response letter to Referee #2 on the manuscript "Monte Carlo method for determining uncertainty of total ozone derived from direct solar irradiance spectra: Application to Izaña results"**

Authors: Anna Vaskuri, Petri Kärhä, Luca Egli, Julian Gröbner, and Erkki Ikonen

Article reference: amt-2017-403

Authors: We thank Anonymous Referee #2 for carefully reading our AMT Discussion paper and giving constructive criticism. We will later answer point by point all details raised, but at this stage we clarify the scientific concerns raised.

**Comments on Introduction**

"First, the paper was probably written very quickly and important information is missing, which makes the understanding quite difficult for the unexperienced reader:

- a complete introduction about the importance of ozone measurements and the assessment of their uncertainty should be included (it is only the first line, so far);

- it is not explained why spectral data should contain correlations (physical basis);"

Authors: We revised Introduction and included new paragraphs as the referee suggested. After these amendments, the introduction is:

[revised manuscript text omitted]

**SPECIFIC COMMENT 1 - Retrieval model**

"It should be explained why the retrieval method (Eq. 8) was chosen. An obvious drawback is that this method is not invariant for spectrally-constant factors ("full correlation" or systematic errors in the absolute calibration, as hypothesised for AVODOR). It should be stated what networks / instruments use this model. For example, it is false that "This approach is consistent with the ozone measurements with Brewer" (p. 7, line 17): the Brewer algorithm is invariant for "full spectral correlation" (therefore, the "full correlation" term would not make sense with a different retrieval method, e.g. for a method where the offset is also included in a DOAS-like fit)."

Authors: Omitting an offset factor in our model was a mistake from our side, as we did not take into account how easily full correlations appear in solar UV irradiance measurements. As the results of AVODOR show, it is clearly needed. We thus modified the atmospheric model as

$$E_\mathrm{s}(\lambda) = c \cdot E_\mathrm{ext}(\lambda) \cdot \exp\bigl[-\alpha_{\mathrm{O}_3}(\lambda, T_\mathrm{eff}) \cdot \mathrm{TOC} \cdot m_\mathrm{TOC} - \tau_\mathrm{R}(\lambda, P_0, z_0, \phi) \cdot m_\mathrm{R} \\ -\tau_\mathrm{AOD}(\lambda) \cdot m_\mathrm{AOD}] \tag{1R}$$

$$\tau_\mathrm{AOD}(\lambda) = \beta \cdot \lambda^{-\alpha}, \tag{2R}$$

where the parameters are as in Eqs. (3)–(7) in the AMT Discussion manuscript and new multiplier $c$ is the scaling factor set as an additional free parameter. Thus, there are three free parameters; TOC, $\beta \geq 0$, and $c$ to be fitted. We set $\beta \geq 0$ since to our understanding aerosols attenuate the direct solar UV spectrum. The offset factor will correct for any wavelength independent deviation in the measurements. The results of the paper change due to this modification. The large offset of AVODOR will diminish. The other devices will also have small changes in their results. We give below discussion of the changed results.

We now call $E_\mathrm{s}(\lambda)$ as the modelled direct spectral irradiance at the Earth surface, and $\beta$ as Ångström turbidity coefficient, although Ångström (1964) himself called $\beta$ as extinction coefficient.

We also studied the effect of using a linear approximation of aerosol optical depth in TOC values estimated as it was used e.g. by Huber et al. (1995). In this test, Eq. (2R) was replaced with

$$\tau_\mathrm{AOD}(\lambda) = a \cdot (\lambda - 340\,\mathrm{nm}) + b, \tag{3R}$$

where $a$ and $b$ are free parameters without bound constraints. When linear AOD model in Eq. (3R) is used with (1R), the multiplier needs to be removed by setting it to $c = 1$, because the free parameter $b$ produces almost similar multiplier ($e^{-b \cdot m_\mathrm{AOD}}$). Based on our analysis, the linear AOD model gave almost similar results as Eq. (2R), so we prefer the Ångström AOD model and an offset factor $c$, as we believe this approach is more physical.

"Also, since this method gives more weight to lower irradiances, the authors should carefully explain how the "wavelength region where the signal is above the noise floor" was determined (5E-3 and 1E-5 noise floors);"

Authors: It is not obvious which least squares minimization method works best, as the method of obtaining TOC from a measured spectrum is very critical to the way that the least squares fitting is performed. We have now studied different options how to perform the weighting. To facilitate discussion, we rewrite Eq. (8) in our AMT Discussion manuscript as

$$S = \sum_{i=1}^{n} w(\lambda_i)[E_\mathrm{s}(\lambda_i) - E(\lambda_i)]^2, \tag{4R}$$

where $w(\lambda_i)$ is the weight.

In our AMT Discussion manuscript, we used relative errors in calculation, i.e., $w(\lambda_i) = E(\lambda_i)^{-2}$. This method was also used in the article by Huber et al. (1995). This method works very well with high-accuracy double-monochromator instruments giving several orders of magnitude of useful data down to $10^{-6}$ Wm$^{-2}$nm$^{-1}$ at UV region, such as the one used in the original study by Huber et al. (1995), or the QASUME used in our experiment. However, with large zenith angles in the morning and in the evening, the spectra have to be cut at some reasonable threshold level even with monochromator-based instruments when the relative least squares minimization is used.

With array spectroradiometers that suffer from stray light and high baseline noise as presented in Fig. 1R, the dynamic range can easily be less than two orders of magnitude. Figure 1R shows the fitted spectra for all three devices, and the residuals of the fits. The residuals show clearly how the stray light is present in both BTS and AVODOR. The threshold level where the data is cut affects the TOC results at large zenith angles significantly, when the relative least squares minimization is used as demonstrated in Figure 2R.

[Figure]

Figure 1R. Examples of fitting the atmospheric model to the direct ground-based solar UV spectra between 300 nm and 340 nm for QASUME (a–b), BTS (c–d) and AVODOR (e–f). In figures on the left hand side, the coloured symbols indicate measured spectra, and the black solid curves indicate modelled spectra. Figures on the right hand side show the relative spectral residuals of the fits. In (a), the abbreviation DR refers to the dynamic range of QASUME data used in the least squares fitting.

[Figure]

Figure 2R. TOC values estimated using different weighting in least squares minimisation and using Ångström AOD model for QASUME (a), BTS (b) and AVODOR (c). TOC values for Brewer #183 are plotted as black crosses for comparison. The colour codes and the associated figure legends denote the weighting used (Eq. (4R)) and the dynamic range DR used.

Response letter to Referee #2
Article reference: amt-2017-403

Figure 2R shows TOC values calculated from spectra throughout the day using various minimization methods. As can be seen, using relative least squares fitting causes significant inverse U-shape to the results obtained with BTS and AVODOR, which is not present in the QASUME data (triangle and square symbols). In the case of BTS and AVODOR, the TOC results change significantly by changing the threshold level. This can be seen in the curves, where the dynamic range of the data was $10^2$ for BTS and $10^{1.5}$ for AVODOR. However, the stray light still causes U-shape to the curves. In our opinion, the most objective method for analysing the results of the array based spectroradiometers is to use absolute least squares minimisation by setting the weight to $w(\lambda) = 1$ (light green and orange circles in Fig. 2R). The absolute least squares minimization does not give much weight to the baseline noise tail. As can be seen in Fig. 2R, with the absolute least squares fitting there is very little variation throughout the day and we do not have to subjectively limit the dynamic range the spectra.

As we know from the results of Brewer and QASUME e.g. in Fig. 2R, the TOC was rather constant during the measurement day. Thus, the inverse U-shape notable for most curves is a property of the analysis method originating from the stray light and the baseline noise that our atmospheric model cannot account for. Herman et al. (2015) have also reported the inverse U-shape in TOC due to stray light of array spectroradiometers.

Our conclusion is that the relative least squares minimization (Eq. (4R) with $w(\lambda) = E(\lambda)^{-2}$) is only applicable to spectroradiometers free from stray light. For our paper, we choose to use this approach to QASUME with the dynamic range of four orders of magnitude. With high-accuracy instruments this is justified, because such weighting uses all data available. For both array spectroradiometers, BTS and AVODOR, we use absolute least squares minimization (Eq. (4R) with $w(\lambda) = 1$). This approach is justified, as it gives less weight to the low irradiance values distorted by stray light.

According to Fig. 2R the array spectroradiometers operate well at noon, even if analysed using relative least squares fitting. Comparing results at noon shows that using absolute least squares fitting underestimates the TOC values by 2 DU as compared to relative least squares fitting. The same offset, independent of zenith angle can be seen with QASUME.

Figure 3R shows the final comparison results. As can be seen, the inverse U-shape has diminished in both BTS and AVODOR data as compared to Fig. 5 of our AMT Discussion paper, and the agreement is rather good for all instruments.

[Figure]

(a) TOC estimated using Eq. (1R) with Ångström AOD model in Eq. (2R)

(b) TOC estimated using Eq. (1R) with linear AOD model in Eq. (3R)

Figure 3R. Absolute TOC values estimated for the spectroradiometers studied. Average of 10 neighbouring values has been included for AVODOR to show the spectral shape behind the noise.

In our revised manuscript, we will include the presented Figures 1R – 3R, and their descriptions. Relative least squares fitting is used with QASUME using the dynamic threshold of four orders of magnitude which can be justified by the extremely low noise level as seen in Figure 1R. For the array spectrometers, we use the absolute least squares fitting. In addition, to avoid confusion we removed the sentence stating that our approach is consistent with Brewer algorithm.

A new reference was included in the manuscript:

Herman J., Evans R., Cede A., Abuhassan N., Petropavlovskikh I., and McConville G.: Comparison of ozone retrievals from the Pandora spectrometer system and Dobson spectrophotometer in Boulder, Colorado, Atmos. Meas. Tech., 8, 3407–3418, 2015.

"For example, it is false that "This approach is consistent with the ozone measurements with Brewer" (p. 7, line 17): the Brewer algorithm is invariant for "full spectral correlation" (therefore, the "full correlation" term would not make sense with a different retrieval method, e.g. for a method where the offset is also included in a DOAS-like fit)."

It is true that with our revised MC uncertainty method, "full correlation" does not contribute, to the uncertainty of TOC. Full correlation in the ozone absorption cross section is an exception to this, since variable $c$ cannot completely compensate for errors in the exponent. However, it is still meaningful to divide the components to full, unfavourable and random correlations. The part that is fully correlated does not contribute to uncertainty, thus the resulting uncertainty will be smaller correspondingly. For example, in Table 4 of our AMT Discussion manuscript, the spectral uncertainty arising from full spectral correlation of the extra-terrestrial spectrum does not contribute to the uncertainty in TOC, but it contributes to the total uncertainty of the extra-terrestrial spectrum.

**SPECIFIC COMMENT 2 - Uncertainty model**

"The uncertainty model (Eq. 9-10) is very complex. However, for the most part of the paper, only three terms of the Fourier series are used. Thus, I wonder why such a complex initial framework must be described."

Authors: The Fourier series approach is intended as a generic tool that can be applied to any quantity derived mathematically from a spectrum. This is explained on lines 19–20 on page 10 of the AMT Discussion manuscript. In the publication by Kärhä et al. (2018), it is demonstrated that many national metrology institutes have systematic wavelength dependent deviations in their spectral irradiance scales. The Fourier series can reproduce such spectral shapes. The main benefit of using the Fourier series is that the variance of the systematic spectral deviation can easily be controlled, as introduced in Section 4.1 of our AMT Discussion manuscript.

An updated reference:

Kärhä P., Vaskuri A., Pulli T., and Ikonen E.: Key comparison CCPR-K1.a as an interlaboratory comparison of correlated color temperature, J. Phys.: Conf. Ser. 972, 012012, 2018. doi:10.1088/1742-6596/972/1/012012

"Also, it should be explained why deviations in measurements should follow this model. Coming to the "unfavourable correlation", "The obtained TOC value is affected most by spectral distortion that mimics the spectral shape of the ozone absorption... The first combination of constant offset and one sinusoidal function ... is closest to this extreme"."

Authors: As it is explained on lines 19–20 on page 10 of the AMT Discussion manuscript, the model does not assume any particular spectral shape of the deviation. Fourier series has a property that in its full form it can mimic any shape of deviation. This takes place at the Nyquist criterion, where N is equal to half the number of wavelengths available. Also with smaller values of N, the MC parameters can account for unknown spectral shapes of lower complexity.

We sum the base functions starting with the simplest forms, N = 1, 2 ... The logic here is that when one studies which kind of correlations there are present in typical measurements, correlations such as these appear commonly. This can be seen in (Kärhä et al. (2018)) and (Woolliams et al. (2006)) which examine deviations in the spectral irradiance measurements of national standards laboratories.

A reference:

Woolliams E. R., Fox N. P., Cox M. G., Harris P. M. and Harrison N. J.: Final report on CCPR-K1-a: Spectral irradiance from 250 nm to 2500 nm, Metrologia, 43, Technical Supplement 02003, 2006.

"However, from Eq. 10, this term varies with lambda1 and lambda2, so the width of the sinusoid is different for the three instruments and comparison of the results is difficult."

Authors: This is true. We now changed our analysis so that all instruments use the same wavelength region 300 nm – 340 nm and remodelled the results.

"Also, why should this term be a full sinusoid cycle? Why should the period of the oscillation be the same for all uncertainty components (calibration, measurement, cross sections, etc.)?"

Authors: As it is explained on lines 22–25 on page 18 of the AMT Discussion manuscript, simpler forms of deviations could be used after the general solution of the problem is known.

In our method, adding half a sinusoid or a slope as a base function in the general solution would lead into more complicated calculations when determining the rest of the base functions, as all the base functions need to be orthogonal with respect to each other. It is possible to use other orthogonal sets of functions instead of sinusoidal functions, but that would involve more complicated mathematics.

"Finally, it is obvious that phi has an enormous role for the "unfavourable correlation" term: depending on phi's value (i.e. similarity with ozone spectral cross section), the induced error could be huge or negligible. The role of phi should be explained better. Physically, what does phi represent? It is expected to randomly change in one instrument or not?"

Authors: Parameter $\phi_i$ is used as an equally distributed MC variable between 0 and $2\pi$ (see page 9 lines 29–30 of the AMT Discussion manuscript). It is needed in the Fourier series so that it can reproduce all possible spectral shapes.

---

## Author Comment (AC2) · 24 Mar 2018

**Response letter to Referee #4 on the manuscript "Monte Carlo method for determining uncertainty of total ozone derived from direct solar irradiance spectra: Application to Izaña results"**

Authors: Anna Vaskuri, Petri Kärhä, Luca Egli, Julian Gröbner, and Erkki Ikonen

Article reference: amt-2017-403

*Authors:* We thank Anonymous Referee #4 for the valuable comments that helped us in improving the manuscript. We have included below our detailed responses to all comments.

**Abstract**

"**P1, L1** replace "calculate" with "estimate"."

> *Authors:* Corrected according to reviewer's suggestion. The new sentence is: "We demonstrate a Monte Carlo model to estimate the uncertainties of total ozone column (TOC), derived from ground-based direct solar spectral irradiance measurements."

**Introduction**

"**P1, L13-14** At this point the authors should make clear that they are talking for correlations in spectral measurements. According to the authors this is the main problem solved when the new methodology is applied. I also suggest adding more information here to help the reader understand what they mean when they refer to "correlations"."

> *Authors:* We revise the text about spectral correlations in the introduction and include a new paragraph with some examples about where they might arise:

> "TOC can be determined from spectral measurements of direct solar UV irradiance (Huber *et al.* (1995)). We have developed a Monte Carlo (MC) based model to estimate the uncertainties of the derived TOC values. One frequently overlooked problem with uncertainty evaluation is that the spectral data may hide systematic wavelength dependent errors due to unknown correlations (Kärhä *et al.* (2017b, 2018); Gardiner *et al.* (1993)). Omitting possible correlations may lead into underestimated uncertainties for derived quantities, since spectrally varying systematic errors typically produce larger deviations than uncorrelated noise-like variations that traditional uncertainty estimations predict. Complete uncertainty budgets for quantities measured are necessary to understand long term environmental trends, such as changes in the stratospheric ozone concentration (e.g. Molina and Rowland (1974)) and solar UV radiation (e.g. Kerr and McElroy (1993); McKenzie *et al.* (2007)).

> Physically, correlations may originate, e.g., from lamps or other light sources used in calibrations. If their temperatures change e.g. due to ageing or current setting, a spectral change in the form of Planck's radiation law is introduced. Non-linearity in the responsivity of a detector causes systematic differences between high and low measured values. The introduced spectrally systematic but unknown changes in irradiance may change the derived TOC values significantly, exceeding the uncertainties calculated assuming that the uncertainty

in irradiance behaves like noise. The presence of correlations in measurements can be seen in many ways. For example, problems have occurred when new ozone absorption cross-sections have been taken into use (Redondas *et al.* (2014); Fragkos *et al.* (2015)). Derived ozone values may change significantly because different systematic errors are included in the different cross-sections. Also, TOC estimated from a measured spectrum often depends on the wavelength region chosen, although the measurement region should not affect the result much."

Regarding these paragraphs, a new reference is included in the manuscript and one reference is updated:

Redondas A., Evans R., Stuebi R., Köhler U., and Weber M.: Evaluation of the use of five laboratory-determined ozone absorption cross sections in Brewer and Dobson retrieval algorithms, Atmos. Chem. Phys., 14, 1635–1648, 2014.

Kärhä P., Vaskuri A., Pulli T., and Ikonen E.: Key comparison CCPR-K1.a as an interlaboratory comparison of correlated color temperature, J. Phys.: Conf. Ser., 972, 012012, 2018. doi:10.1088/1742-6596/972/1/012012

**"P2, L4-5** Do you mean here that the field of view of the spectroradiometers is equal to exactly one solar diameter? If not, then some scattered irradiance also enters the spectrometer."

*Authors:* The field of view of each spectroradiometer was limited. We revise the sentence as: "The field of view of the spectroradiometers has been limited so that they measure direct spectral irradiance of the Sun, excluding most of the indirect radiation from the remainder of the sky."

We also rename Section 2 as "ATMOZ field measurement campaign and instrument description" and included the field of view of each spectroradiometer in Section 2. The field of view with a full opening angle is 2.5° for QASUME (Gröbner *et al.* (2017)), 2.8° for BTS (Zuber *et al.* (2017b)), and 1.5° for AVODOR according to the manual of the collimator tube used, J1004-SMA by CMS Ing.Dr.Schreder GmbH.

**Section 2**

**"The tables 1, 2 and 3** are presented here without any discussion regarding the presented quantities. I suggest that they should be moved to the uncertainty estimation section (section 4). Furthermore, some discussion (e.g. explaining the presented correlation types, description of how the different uncertainty types were estimated) would be useful."

*Authors:* We admit these tables are better suited in Section 4, after the spectral correlation types have been introduced. We move measurement uncertainty tables for QASUME, BTS, and AVODOR (old Tables 1, 2 and 3) to Section 4. We also include more discussion about the uncertainty components:

"The uncertainties due to radiometric calibration include factors such as the uncertainty of the standard lamp used, and the additional uncertainty due to noise and alignment. QASUME has been validated using various methods, thus the uncertainty due to calibration is low (Hülsen *et al.* 2016). For QASUME and BTS, we assume the correlations to be equally distributed between full correlation, unfavourable correlation, and random correlation (Kärhä *et al.* 2018). Spectra measured with AVODOR are significantly noisier, thus half of the uncertainty is associated to the random component. Values for instability of the calibration lamp are based

on long-term monitoring. The lamp irradiances have been noted to gradually drop with no significant wavelength structure within the wavelength region concerned. Nonlinearity values are estimations of the operators of the devices. Nonlinearity is typically manifested so that the responsivity of the device changes gradually from high readings to low readings. This can cause significant change in the TOC values, thus we assume the correlation type to be unfavourable. Uncertainties due to device stability and temperature dependence are based on long-term monitoring. The changes have been found to be independent on wavelength in the region concerned, thus full correlation is assumed. Noise is the average standard deviation of typical measurements at noon over the wavelength region concerned. The wavelength scales of the devices have been checked using emission lines of gas discharge lamps. The uncertainty values given are the estimated standard deviations of the possible remaining errors after corrections. Wavelength error can introduce a significant change in TOC, because it introduces an error in the form of the derivative of the spectral irradiance. Thus, unfavourable correlation is assumed. Most of the uncertainty components are slightly wavelength dependent but to simplify simulations, average uncertainty values are used over the wavelength range between 300 nm and 340 nm."

**Section 3**

**"P7, L2** Gröbner and Kerr (2001) did not assume that the air mass factors for aerosols and Rayleigh scattering are equal."

*Authors:* Indeed, Gröbner and Kerr (2001) did not deal with aerosols. The assumption was taken from the paper by Gröbner *et al.* (2017). We revise the sentence as: "As the ozone and other molecules creating scattering are distributed at different altitudes, we calculate the relative air mass factor $m_R$ for Rayleigh scattering at the altitude of 5 km (Gröbner and Kerr (2001)) and approximate the effective altitude of aerosols so that $m_{AOD} \approx m_R$ (Gröbner *et al.* (2017))."

References:

Gröbner J. and Kerr J. B.: Ground-based determination of the spectral ultraviolet extraterrestrial solar irradiance: Providing a link between space-based and ground-based solar UV measurements, J. Geophys. Res., 106, 7211–7217, 2001.

Gröbner J., Kröger I., Egli L., Hülsen G., Riechelmann S., and Sperfeld P.: The high resolution extra-terrestrial solar spectrum determined from ground-based solar irradiance measurements, Atmos. Meas. Tech., 10, 3375–3383, 2017.

**Section 4**

**"P14, L5** Again, the reference of Gröbner and Kerr (2001) is not correct here."

*Authors:* We change the reference and revise the sentence as: "Rayleigh scattering and aerosols are set at the altitude of 5 km ± 0.5 km, which influences the relative air mass $m_R \approx m_{AOD}$ (Gröbner *et al.* (2017))."

**Section 5**

**"P17** Please add more information regarding the linear model used for AOD. E.g., why using the particular model for AOD? Are a and b the same with those of Ångström (1964)? If not, how they are estimated? What happens if the TOC is derived by QASUME and BTS using this linear AOD model?"

**Authors:** For example, Huber *et al.* (1995) use such a linear model for AOD. The aerosol model by Ångström (1964) can be approximated with a line when a narrow spectral range is modelled. In the linearized AOD model, parameters $a$ and $b$ do not have exact physical meanings, they are just coefficients. Mostly, because of this, we choose to use the model by Ångström (1964), as it is more physical, but we also compare some of our results with results obtained using the linear equation.

In response to the comments by Anonymous Referee #2, we include an offset factor $c$ to the atmospheric model of Eq. (3) of our AMT Discussion manuscript to compensate for full spectral correlations as:

$$E_s(\lambda) = c \cdot E_{\text{ext}}(\lambda) \cdot \exp\left[-\alpha_{O_3}(\lambda, T_{\text{eff}}) \cdot \text{TOC} \cdot m_{\text{TOC}} - \tau_R(\lambda, P_0, z_0, \phi) \cdot m_R\right.$$
$$\left. -\tau_{\text{AOD}}(\lambda) \cdot m_{\text{AOD}}\right]. \tag{1R}$$

After this change, the atmospheric model has three free fitting parameters: TOC, $\beta \geq 0$, and $c$, and the TOC estimated for AVODOR at local noon agrees quite well with other instruments. There is still the inverse U-shape in the BTS and AVODOR results, but it diminishes when the relative least squares fitting

$$S = \sum_{i=0}^{N} w_i \left[E_s(\lambda_i) - E(\lambda_i)\right]^2 \tag{2R}$$

with $w(\lambda) = E(\lambda)^{-2}$ is replaced with the least squares fitting of absolute residuals by setting $w(\lambda) = 1$.

To justify our approach, we compare in Fig. 1R the results obtained using Eq. (1R) with the Ångström AOD model of Eq. (7), to those obtained using Eq. (1R) with the linear AOD model of Eq. (13). When we used linear AOD model, we kept all parameters $a$, $b$ and TOC as free fitting parameters without bound constraints. In addition, we set $c = 1$, because $b$ produces an almost similar offset factor ($e^{-b \cdot m_{\text{AOD}}}$) as $c$. Parameter $a$ compensates for slope-like spectral deviations.

According to the new simulations presented in Fig. 1R, the TOC values obtained using the linear AOD model are practically the same for BTS and AVODOR as those obtained using the Ångström model and an offset factor $c$. With QASUME, the results deviate by 2 DU due to the unconstrained slope factor $a$ of the linear AOD model.

[Figure]

*Figure 1R. Absolute TOC values estimated for the spectroradiometers studied. Average of 10 neighbouring values has been included for AVODOR to show the spectral shape behind the noise. Abbreviation DR refers to the dynamic range of QASUME data used in the least squares fitting.*

**Conclusions**

**"P18, L27** The results from AVODOR deviate up to 10 DU (and not 10 DU) depending on the SZA. Is the stray light effect enough to explain these discrepancies?"

> *Authors:* There is a spectrally constant offset in the spectral irradiance measured by AVODOR that our model in the AMT Discussion manuscript could not handle. We did not take into account how easily full correlations appear in solar UV irradiance measurements, e.g., due to geometrical factors. Thus, we improved our atmospheric model by including an offset factor $c$ as a free parameter in Eq. (1R). Stray light is mostly responsible for the inverse U-shape of TOC (Herman *et al.* (2015)), but using absolute least squares fitting, i.e., by setting the weight to $w(\lambda) = 1$ in Eq. (2R), we get rid of the solar zenith angle dependence. The TOC estimated from the spectra of all the instruments with the improved atmospheric model are presented in Fig. 1R.
>
> We include a new reference in the manuscript:
>
> Herman J., Evans R., Cede A., Abuhassan N., Petropavlovskikh I., and McConville G., "Comparison of ozone retrievals from the Pandora spectrometer system and Dobson spectrophotometer in Boulder, Colorado," Atmos. Meas. Tech., 8, 3407–3418, 2015.

**"The last paragraph** of the conclusions section is now written leads to the conclusion that the main outcome of the study is that the AVODOR is not suitable for TOC measurements, while QASUME and BTS are. In my opinion the main outcome of this study is that the presented method provides more accurate estimations of the uncertainty budget compared to the traditionally used methods. However, it is not adequate for properly estimating uncertainties if the instruments are not characterized for systematic measurement errors. I suggest re-writing the conclusions section in a way that the main conclusions of the study are highlighted."

> *Authors:* It is true that the conclusions in its present form give too much weight to the comparison of the devices. We will rewrite the conclusions to give more emphasis to the correlation issues. It is worth noting that in response to Referee #2, we revise the algorithm

for obtaining TOC from spectra (Please, see separate response letter addressed to Referee #2 for full details). The least squares fitting is modified so that the low irradiance values distorted by stray light with BTS and AVODOR get less weight, and the offset of AVODOR gets corrected. After this change, the results are in better agreement, and the daily variation of TOC seen with BTS and AVODOR diminishes (Please, see new results in Fig. 1R). AVODOR seems to work better than first expected. The results are just noisy but quite well in agreement with other devices. We will write the new conclusions after going through all Referee comments. We need to include some discussion about the model change into the conclusions as well, but we try to keep the emphasis on the uncertainty issue.

**Additional notes by the authors**

As we modified the retrieval algorithm by including a new offset factor $c$ to compensate for full spectral deviations, the results compared to the AMT Discussion paper will change. We will replace Fig. 5 in the AMT Discussion manuscript with Fig. 1R shown in this document.

---

## Author Comment (AC3) · 26 Mar 2018

**Response letter to Referee #3 on the manuscript "Monte Carlo method for determining uncertainty of total ozone derived from direct solar irradiance spectra: Application to Izaña results"**

Authors: Anna Vaskuri, Petri Kärhä, Luca Egli, Julian Gröbner, and Erkki Ikonen

Article reference: amt-2017-403

Authors: We thank Anonymous Referee #3 for the valuable comments that helped us in improving the manuscript. We have included below our detailed responses to all comments.

**Specific comments**

"P.4. Table 1. – 3. The table for Uncertainties including the fraction of correlation need further explaining, especially since the issue of correlation is introduced much later in section 4. If these uncertainties have been published in that form earlier, quotation would be helpful in the figure caption. If not, these tables should be moved to section 4 prior to table 4 or in combination with table 4 since the explaining is done on p.15"

Authors: We agree, and move Tables 1, 2 and 3 to Section 4. Most of the uncertainties are estimated for this manuscript or obtained from personal communication, and they have not been published elsewhere. We include a new paragraph in Section 4 to explain how the uncertainties are obtained:

"The uncertainties due to radiometric calibration include factors such as the uncertainty of the standard lamp used, and the additional uncertainty due to noise and alignment. QASUME has been validated using various methods, thus the uncertainty due to calibration is low (Hülsen et al. (2016)). For QASUME and BTS, we assume the correlations to be equally distributed between full correlation, unfavourable correlation, and random correlation (Kärhä et al. (2018)). Spectra measured with AVODOR are significantly noisier, thus half of the uncertainty is associated to the random component. Values for instability of the calibration lamp are based on long-term monitoring. The lamp irradiances have been noted to gradually drop with no significant wavelength structure within the wavelength region concerned. Non-linearity values are estimations of the operators of the devices. Non-linearity is typically manifested so that the responsivity of the device changes gradually from high readings to low readings. This can cause significant change in the TOC values, thus we assume the correlation type to be unfavourable. Uncertainties due to device stability and temperature dependence are based on long-term monitoring. The changes have been found to be independent on wavelength in the region concerned, thus full correlation is assumed. Noise is the average standard deviation of typical measurements at noon over the wavelength region concerned. The wavelength scales of the devices have been checked using emission lines of gas discharge lamps. The uncertainty values given are the estimated standard deviations of the possible remaining errors after corrections. Wavelength error can introduce a significant change in TOC, because it introduces an error in the form of the derivative of the spectral irradiance. Thus, unfavourable correlation is assumed. Most of the uncertainty components are slightly wavelength dependent but to simplify

simulations, average uncertainty values are used over the wavelength range between 300 nm and 340 nm."

A reference is updated:

Kärhä P., Vaskuri A., Pulli T., and Ikonen E.: Key comparison CCPR-K1.a as an interlaboratory comparison of correlated color temperature, J. Phys.: Conf. Ser., 972, 012012, 2018. doi:10.1088/1742-6596/972/1/012012

"L4-8. Why is uncertainty of radiometric calibration of AVODOR so much higher than for the other instruments? Just because of low SNR in the UV region?"

Authors: Yes, the uncertainty is higher mainly because of the noise. AVODOR response is noisier and less stable as compared with QASUME and BTS.

"The stated uncertainties given in the tables are valid for the whole spectral range of the instrument? I would expect uncertainties related to radiometric calibration and measurement noise to be wavelength dependent. Or are these stated values the upper limit of uncertainties? In the following text, there is a lot discussion about straylight effects. However, there is no explicit uncertainty component related to straylight or straylight correction?"

Authors: It is true that the uncertainties and noise levels are slightly dependent on the wavelength. We simplified the simulations by using average values. A sentence on this is included to the text in Section 4: "Most of the uncertainty components are slightly wavelength dependent but to simplify simulations, average uncertainty values are used over the wavelength range between 300 nm and 340 nm."

Stray light level is difficult to estimate, and to calculate its effect on the TOC uncertainties, we would need the stray light matrix for each instrument (Nevas et al. (2014)) that we do not have. A more practical way of obtaining its effect is to compare the results with a device that does not suffer from stray light. In this case, QASUME is significantly better with respect to stray light than the two array spectroradiometers, BTS and AVODOR. We include discussion about this in the paper.

In the case of array spectroradiometers, the stray light and baseline noise severely affect the TOC analysis at large zenith angles when the fitting is performed with the relative least squares method. This can be seen in Fig. 1R as an inverse U-shape of TOC values determined from BTS and AVODOR spectra when the relative least squares minimisation

$$S = \sum_{i=1}^{n} w(\lambda_i)[E_s(\lambda_i) - E(\lambda_i)]^2 \qquad (1R)$$

with $w(\lambda) = E(\lambda)^{-2}$ is used. If the least squares method is modified so that absolute residuals ($w(\lambda) = 1$) are minimised, the inverse U-shape of TOC results at large zenith angles diminishes. With the absolute least squares fitting, the TOC values at noon are 2 DU lower as compared with those estimated using the relative least squares fitting. We include this analysis and Fig. 1R in the revised manuscript. This issue is discussed in more detail in our response to Referee #2.

[Figure]

Figure 1R. TOC values estimated using different weighting in least squares minimisation and using Ångström AOD model for QASUME (a), BTS (b) and AVODOR (c). TOC values for Brewer #183 are plotted as black crosses for comparison. The colour codes and the associated figure legends denote the weighting used and the dynamic range DR used.

Reference:

Nevas S., Gröbner J., Egli L., and Blumthaler M.: Stray light correction of array spectroradiometers for solar UV measurements, Appl. Opt., 53, 4313–4319, 2014.

"P. 7, L15 The least square fitting might lead to local minima instead of the global minimum. How is that accounted for? Especially if AOD and TOC are both fitting parameters."

> Authors: We performed the least squares fitting using a Matlab function 'lsqnonlin'. We set optimoptions(@lsqnonlin, 'Algorithm', 'trust-region-reflective', 'MaxFunEvals', 1000, 'MaxIter', 1000, 'StepTolerance', 1e-10, 'OptimalityTolerance', 1e-10, 'FunctionTolerance', 1e-10). In our case, one of these tolerances was met by less than 50 iterations when the initial guess values were set to $TOC = 200$, $\beta = 0$, and $c = 1$.

> We included an offset factor $c$ as free fitting parameter in front of $E_{\text{ext}}(\lambda)$ as suggested by Referee #2. After this change, we have three parameters (TOC, $\beta$, $c$) to be fitted. We also set $\beta \geq 0$ using the bound constraints of 'lsqnonlin' function, as aerosols attenuate solar UV spectrum.

> To our understanding, global minimum should be achieved quite easily with two or three free parameters, but of course, it depends on algorithms used and the data set to be fitted. We tested the robustness of our retrieval by varying the initial guess values of TOC, $\beta$, and $c$. We varied the initial TOC value between 10 DU – 700 DU, the initial guess value of $\beta$ between 0 – 0.5, and the initial guess value of $c$ between 0 – 100. Using the initial guess values within such ranges, the free parameters always converged to the same final values and they were independent on the initial guess values. We include text on these tests in the AMT Discussion manuscript on page 7 after line 16.

"P.9, L20 "In this paper, we do this for all components, where the mechanism of contributing to the uncertainty of TOC is known." I guess these components are those with correlation "full" and "random"? This could be specified here."

> Authors: Actually, the text refers to those components, where fractions for correlations are not listed in Table 4. Instead, they are labelled as (a) – (d). We clarify this by including a sentence before line 5 on page 14:

> "For components (a) – (d) in Table 4, the mechanism of contributing to the uncertainty of TOC is known. We know the standard uncertainty of the $O_3$ layer altitude of 26 km to be u = 0.5 km, so we vary the altitude accordingly and note the variance of the resulting TOC."

"P.11, Figure 4 That figure is a bit confusing. For underlining the statement on full, unfavorable and random correlation the display of one colored graph is sufficient. The additional information gained from the u=5% and u=2.5% graph, as well as the black solid lines is not explained in the plain text and incomprehensible explained in the figure caption."

> Authors: All those curves were intended to show the scalability of the model, but we agree that they were not sufficiently explained in the text. To avoid confusion, we replace Fig. 4 of our AMT Discussion manuscript with Fig. 2R shown below. The new Fig. 4 includes the values obtained by varying the spectral irradiance (circles) and also the values obtained by varying the ozone absorption cross section (crosses). The uncertainty behaves differently if the parameter to be analysed is in front of the equation as a multiplier (such as spectral irradiance) or in the exponent as a direct multiplier to TOC (such as the $O_3$ cross section). We include a new sentence on page 10, line 16, of the AMT Discussion manuscript as:

> "The uncertainty in TOC arising from the spectral deviation in $\alpha_{O_3}(\lambda, T_{\text{eff}})$ is plotted as crosses in Fig. 4 as a function of increasing N."

[Figure]

Figure 2R. Standard uncertainties of TOC at local noon as a function of the order of complexity N for QASUME spectroradiometer, with 1% standard deviation in spectral irradiance $E(\lambda)$ plotted as circles, and 1% standard deviation in ozone absorption cross-section $\alpha_{O_3}(\lambda, T_{\text{eff}})$ plotted as crosses.

**Additional notes by the authors**

As we modified the retrieval algorithm by including a new offset factor $c$ to compensate for full spectral deviations, the results as compared to the AMT Discussion paper will slightly change. Please, see separate response letter addressed to Referee #2 for full details.

---

## Author Comment (AC4) · 29 Mar 2018

**Response letter to Referee #1 on the manuscript "Monte Carlo method for determining uncertainty of total ozone derived from direct solar irradiance spectra: Application to Izaña results"**

Authors: Anna Vaskuri, Petri Kärhä, Luca Egli, Julian Gröbner, and Erkki Ikonen

Article reference: amt-2017-403

*Authors:* We thank Anonymous Referee #1 for the valuable comments that helped us in improving the manuscript. We have included below our detailed responses to all comments.

**Specific Comments**

"**P1 L14:** "The reason often is that the correlations are not known". It would be better to insert "unknown" into the previous sentence before "correlations" and remove this sentence entirely."

> *Authors:* We remove that sentence and move "unknown" to the sentence before. After this change, the sentence is: "One frequently overlooked problem with uncertainty evaluation is that the spectral data may hide systematic wavelength dependent errors due to unknown correlations (Kärhä *et al.* (2017b, 2018); Gardiner *et al.* (1993))."

"**P1 L17:** This would be better phrased as a complete uncertainty budget being necessary to understand long term environmental trends, rather than increased uncertainties improving the reliability of long term trends."

> *Authors:* We revise the sentence as: "Complete uncertainty budgets for quantities measured are necessary to understand long term environmental trends, such as changes in the stratospheric ozone concentration (e.g. Molina and Rowland (1974)) and solar UV radiation (e.g. Kerr and McElroy (1993); McKenzie *et al.* (2007))."

"**P2 L5:** Better as "...excluding the remainder of the sky". Also should state field of view of each instrument here or a later point in the field campaign section."

> *Authors:* We revise the sentence as: "The field of view of the spectroradiometers has been limited so that they measure direct spectral irradiance of the Sun, excluding most of the indirect radiation from the remainder of the sky."

> We included the field of view of each spectroradiometer in Section 2: The field of view with a full opening angle is 2.5° for QASUME (Gröbner *et al.* (2017)), 2.8° for BTS (Zuber *et al.* (2017b)), and 1.5° for AVODOR according to the manual of the collimator tube used, J1004-SMA by CMS Ing.Dr.Schreder GmbH.

"**P2 L6:** This section may be better called "The Izaña field measurement campaign and instrument description", and include some additional details on each instrument - e.g. the field of view and other pertinent details. Alternatively the uncertainty tables would be better moved to later in the manuscript where the individual contributions are discussed."

*Authors:* We revise the title of Section 2 as: "ATMOZ field measurement campaign and instrument description" and move Tables 1, 2 and 3 to Section 4.

We include more details of the instruments after line 10 on page 3 as:

"The data sets measured by three different spectroradiometers were studied in this work. These spectroradiometers use different techniques to measure the spectral distribution of radiation. Monochromator-based spectroradiometers like QASUME, measure one nearly monochromatic wavelength band at a time, and thus measuring the full spectrum is relatively slow. On the other hand, they usually have significantly better stray light properties than array-based spectroradiometers like BTS and AVODOR, which image the full spectrum at once by dispersing the incoming radiation towards a photodiode array.

QASUME spectroradiometer collects and guides the incoming radiation with input optics and a quartz fiber bundle to the entrance slit of a Bentham DM150 double monochromator (Gröbner *et al.* (2005)). One wavelength at a time can be selected by rotating the two gratings of the double-monochmomator. Then, the monochromatic signal is measured with a photomultiplier tube. QASUME is usually operated in global spectral irradiance mode (Gröbner *et al.* (2005); Hülsen *et al.* (2016)), but during the campaign it was equipped with a collimator tube with a full opening angle of 2.5° allowing the measurement of direct solar spectral irradiance (Gröbner *et al.* (2017)). The measurement range of QASUME during the campaign was limited to 250 nm – 500 nm with a step interval of 0.25 nm, so that one spectrum was measured every 15 minutes. To ensure stable outdoor measurements, the double-monochromator of QASUME was mounted inside a temperature-controlled weather-proofed box (Hülsen *et al.* (2016)).

BTS spectroradiometer utilizes a stationary grating and a back-thinned cooled CCD array detector, mounted in a Czerny-Turner configuration (Zuber *et al.* (2017a, b)). To measure direct solar spectrum, BTS was equipped with a collimator tube with a full opening angle of 2.8° designed by PTB, and it uses an internal filter wheel system with 8 filter positions together with a specific measurement routine to reduce stray light. BTS was mounted on a solar tracker, EKO STR-32G by EKO Instruments Co., Ltd., with pointing accuracy better than 0.01°. A weather-proof housing with temperature control allows BTS operation at the ambient temperatures from –25 °C to 50 °C. During the ATMOZ campaign, the housing temperature of BTS was measured to be stable within 0.1 °C (Zuber *et al.* (2017b)). The measurement range of BTS was 200 nm – 430 nm with a step size of 0.2 nm during the campaign. One spectrum was measured every 45 seconds.

AVODOR spectroradiometer has a stationary grating and a back-thinned cooled CCD array detector in a Czerny-Turner configuration. AVODOR measures the spectrum from 200 nm to 540 nm with a step size of 0.14 nm in the UV region. During the ATMOZ campaign, the field of view of AVODOR was limited to 1.5° by a commercial collimator tube used, J1004-SMA by CMS Ing.Dr.Schreder GmbH. The spectral range of AVODOR was limited between 295 nm and 345 nm

by a combination of two solar blind filters to reduce stray light from the visible and infrared parts of the solar spectrum. The solar blind filters were mounted between the collimator tube and the fiber entrance of the spectroradiometer. One spectrum was measured every 30 seconds."

"**P2 L15:** "mountain Teide ...." » "Mount Teide at *an* altitude...""

*Authors:* Corrected according to reviewer's suggestion. The revised sentence reads: "The measurements took place on the Mount Teide at an altitude of 2.36 km above the sea level (28.3090° N, 16.4990° W)."

"**P2 L16** "Station pressure of 772.8 hPa was monitored during the campaign with a standard uncertainty of 1.3 hPa" » "Station pressure was monitored during the campaign and determined to be 772.8 hPa with a standard uncertainty of 1.3 hPa""

*Authors:* Corrected according to reviewer's suggestion.

"**P4 L4:** "The tables also give division of the uncertainty components to different correlation types as described in Section 4." » "The tables also attribute uncertainty contributions to different correlation types as described in Section 4." I think this is what you mean, either way needs a rephrasing to clarify."

*Authors:* We admit this sentence is not clear so we remove it. We move Tables 1, 2 and 3 to Section 4.

"**P10 L2:** "equal *to* unity""

*Authors:* Corrected according to reviewer's suggestion. The revised sentence is: "The weights $\gamma_i$ for the base functions are selected in an *N*-dimensional spherical coordinate system (Hicks and Wheeling (1959)) in such a way that the variance of the final deviation function equals to unity."

"**P10:** This section on the MC description is clearly the core of the study and where the error estimates are derived, but needs more work and clarification. The details and reasoning behind the approach may be in Karha et al 2017, but it would assist reader of this manuscript to relate MC model and, for example, its sinusoidal terms to physical sources of uncertainty, and how these are calculated for random, unfavourable, and full correlations. At present this isn't clear."

*Authors:* We agree that the core of the paper needs more attention, and we clarify many parts in the text. The text about spectral correlations in the introduction is modified as:

"TOC can be determined from spectral measurements of direct solar UV irradiance (Huber *et al.* (1995)). We have developed a Monte Carlo (MC) based model to estimate the uncertainties of the derived TOC values. One frequently overlooked problem with uncertainty evaluation is that the spectral data may hide systematic wavelength dependent errors due to unknown correlations (Kärhä *et al.* (2017b, 2018); Gardiner *et al.* (1993)). Omitting possible correlations may lead into underestimated uncertainties for derived quantities, since spectrally varying systematic errors typically produce larger deviations than uncorrelated noise-like variations that traditional uncertainty estimations predict. Complete uncertainty budgets for quantities measured are necessary to understand long term environmental trends, such as changes in the stratospheric

ozone concentration (e.g. Molina and Rowland (1974)) and solar UV radiation (e.g. Kerr and McElroy (1993); McKenzie *et al.* (2007)).

Physically, correlations may originate, e.g., from lamps or other light sources used in calibrations. If their temperatures change e.g. due to ageing or current setting, a spectral change in the form of Planck's radiation law is introduced. Non-linearity in the responsivity of a detector causes systematic differences between high and low measured values. The introduced spectrally systematic but unknown changes in irradiance may change the derived TOC values significantly, exceeding the uncertainties calculated assuming that the uncertainty in irradiance behaves like noise. The presence of correlations in measurements can be seen in many ways. For example, problems have occurred when new ozone absorption cross-sections have been taken into use (Redondas *et al.* (2014); Fragkos *et al.* (2015)). Derived ozone values may change significantly because different systematic errors are included in the different cross-sections. Also, TOC estimated from a measured spectrum often depends on the wavelength region chosen, although the measurement region should not affect the result much."

Regarding the above paragraphs, a new reference was included in the AMT Discussion manuscript:

Redondas A., Evans R., Stuebi R., Köhler U., and Weber M.: Evaluation of the use of five laboratory-determined ozone absorption cross sections in Brewer and Dobson retrieval algorithms, Atmos. Chem. Phys., 14, 1635–1648, 2014.

Near Tables 1 – 3 about the uncertainty budgets of the three spectroradiometers, which were moved to Section 4, we place text about how the uncertainties and correlations have been estimated:

"The uncertainties due to radiometric calibration include factors such as the uncertainty of the standard lamp used, and the additional uncertainty due to noise and alignment. QASUME has been validated using various methods, thus the uncertainty due to calibration is low (Hülsen *et al.* (2016)). For QASUME and BTS, we assume the correlations to be equally distributed between *full* correlation, *unfavourable* correlation, and *random* correlation (Kärhä *et al.* (2018)). Spectra measured with AVODOR are significantly noisier, thus half of the uncertainty is associated to the *random* component. Values for instability of the calibration lamp are based on long-term monitoring. The lamp irradiances have been noted to gradually drop with no significant wavelength dependent structure within the wavelength region concerned. Non-linearity values are estimations of the operators of the devices. Non-linearity is typically manifested so that the responsivity of the device changes gradually from high readings to low readings. This can cause significant change in the TOC values, thus we assume the correlation type to be *unfavourable*. Uncertainties due to device stability and temperature dependence are based on long-term monitoring. The changes have been found to be independent on wavelength in the region concerned, thus *full* correlation is assumed. Noise is the average standard deviation of typical measurements at noon over the wavelength region concerned. The wavelength scales of the devices have been checked using emission lines of gas discharge lamps. The uncertainty values

given are the estimated standard deviations of the possible remaining errors after corrections. Wavelength error can introduce a significant change in TOC, because it introduces an error in the form of the derivative of the spectral irradiance. Thus, *unfavourable* correlation is assumed. Most of the uncertainty components are slightly wavelength dependent but to simplify simulations, average uncertainty values are used over the wavelength range between 300 nm and 340 nm."

Regarding the paragraph above, one reference was updated in the AMT Discussion manuscript:

Kärhä P., Vaskuri A., Pulli T., and Ikonen E.: Key comparison CCPR-K1.a as an interlaboratory comparison of correlated color temperature, J. Phys.: Conf. Ser., 972, 012012, 2018. doi:10.1088/1742-6596/972/1/012012

To clarify the uncertainty components in Table 4, we also include new sentences in the AMT Discussion manuscript before line 5 on page 14:

"For components (a) – (d) in Table 4, the mechanism of contributing to the uncertainty of TOC is known. We know the standard uncertainty of the $O_3$ layer altitude of 26 km to be $u$ = 0.5 km, so we vary the altitude accordingly and note the variance of the resulting TOC."

"**P16 L3:** If Brewer #183 is included as a reference instrument, then it would be useful to include a similar uncertainty budget for this instrument, even if only summarised. Also the community would find it useful to put these results into context and comparison with those observed at instrument intercomparisons, and often quoted as a measure of instrument or data quality. i.e. for Brewers is the actual uncertainty determined by the MC methodology much large than expected from the intercomparison error, and, what is the primary source (so efforts can be made to reduce it)."

*Authors:* We thank the referee for the suggestion to include an overall uncertainty budget of Brewer #183. However, this study focuses on MC uncertainty calculation from full spectrum ozone retrieval. An overall uncertainty budget of the double ratio technique ozone retrieval (e.g. for Brewers and Dobsons) is subject of another publication from the ATMOZ field measurement campaign, which is under preparation, but not citable yet. The Brewer data shown in this publication serve only for comparison of the retrieved ozone during the campaign.

**Additional notes by the authors**

As we modified the retrieval algorithm by including a new offset factor $c$ to compensate for full spectral deviations, the results compared with the AMT Discussion paper will slightly change. Please, see separate response letter addressed to Referee #2 for full details.

Response letter to Referee #1
Article reference: amt-2017-403

---

## Author Comment (AC5) · 30 Mar 2018

**Second response letter to Referee #2 on the manuscript "Monte Carlo method for determining uncertainty of total ozone derived from direct solar irradiance spectra: Application to Izaña results"**

Authors: Anna Vaskuri, Petri Kärhä, Luca Egli, Julian Gröbner, and Erkki Ikonen

Article reference: amt-2017-403

*Authors:* We thank Anonymous Referee #2 for the valuable comments that helped us in improving the manuscript. We have included below our detailed responses to the remaining comments. Our response to the comments regarding the introduction and the specific comments 1 and 2 can be found in our first response letter addressed to Referee #2.

**COMMENTS**

"the only characteristics of the instruments described in the manuscript are their spectral range. I believe that a study of the instrumental uncertainties should provide a thorough description of the instruments"

> *Authors:* To provide more information on the campaign and better description of the instruments, we rename Section 2 as "ATMOZ field measurement campaign and instrument description". The following paragraphs after line 10 on page 3 now describe the instruments used:
>
> "The data sets measured by three different spectroradiometers were studied in this work. These spectroradiometers use different techniques to measure the spectral distribution of radiation. Monochromator-based spectroradiometers like QASUME, measure one nearly monochromatic wavelength band at a time, and thus measuring the full spectrum is relatively slow. On the other hand, they usually have significantly better stray light properties than array-based spectroradiometers like BTS and AVODOR, which image the full spectrum at once by dispersing the incoming radiation towards a photodiode array.
>
> QASUME spectroradiometer collects and guides the incoming radiation with input optics and a quartz fiber bundle to the entrance slit of a Bentham DM150 double monochromator (Gröbner *et al.* (2005)). One wavelength at a time is selected by rotating the two gratings of the double-monochmomator. Then, the monochromatic signal is measured with a photomultiplier tube. QASUME is usually operated in global spectral irradiance mode (Gröbner *et al.* (2005); Hülsen *et al.* (2016)), but during the campaign it was equipped with a collimator tube with a full opening angle of 2.5° allowing the measurement of direct solar spectral irradiance (Gröbner *et al.* (2017)). The measurement range of QASUME during the campaign was limited to 250 nm – 500 nm with a step interval of 0.25 nm, so that one spectrum was measured every 15 minutes. To ensure stable outdoor measurements, the double-monochromator of QASUME was mounted inside a temperature-controlled weather-proofed box (Hülsen *et al.* (2016)).

BTS spectroradiometer utilizes a stationary grating and a back-thinned cooled CCD array detector, mounted in a Czerny-Turner configuration (Zuber *et al.* (2017a, b)). To measure direct solar spectrum, BTS was equipped with a collimator tube with a full opening angle of 2.8° designed by PTB, and it uses an internal filter wheel system with 8 filter positions together with a specific measurement routine to reduce stray light. BTS was mounted on a solar tracker, EKO STR-32G by EKO Instruments Co., Ltd., with pointing accuracy better than 0.01°. A weather-proof housing with temperature control allows BTS operation at the ambient temperatures from –25 °C to 50 °C. During the ATMOZ campaign, the housing temperature of BTS was measured to be stable within 0.1 °C (Zuber *et al.* (2017b)). The measurement range of BTS was 200 nm – 430 nm with a step size of 0.2 nm during the campaign. One spectrum was measured every 45 seconds.

AVODOR spectroradiometer has a stationary grating and a back-thinned cooled CCD array detector in a Czerny-Turner configuration. AVODOR measures the spectrum from 200 nm to 540 nm with a step size of 0.14 nm in the UV region. During the ATMOZ campaign, the field of view of AVODOR was limited to 1.5° by a commercial collimator tube used, J1004-SMA by CMS Ing.Dr.Schreder GmbH. The spectral range of AVODOR was limited between 295 nm and 345 nm by a combination of two solar blind filters to reduce stray light from the visible and infrared parts of the solar spectrum. The solar blind filters were mounted between the collimator tube and the fiber entrance of the spectroradiometer. One spectrum was measured every 30 seconds."

"a Monte carlo model was employed, but important details such as the number of samples that were used, the obtained statistical distribution, etc. are not mentioned;"

*Authors:* We add new sentences after line 9 on page 10:

"Each standard uncertainty of TOC in Fig. 4 was estimated from the MC results obtained by running the TOC retrieval 1000 times so that the phases $\phi_i$ and the weights $\gamma_i$ of the base functions were independent at each round. Retrieved TOC deviations resemble Gaussian distribution when the order of complexity of the deviation function is $N \geq 2$."

"the uncertainty components written in the tables are not properly motivated in Sect. 5. Each number should be accompanied by a clear explanation;"

*Authors:* We amend the text describing Tables 1, 2, and 3 to describe all components included. The tables will be located in chapter 4 in the new manuscript:

"The uncertainties due to radiometric calibration include factors such as the uncertainty of the standard lamp used, and the additional uncertainty due to noise and alignment. QASUME has been validated using various methods, thus the uncertainty due to calibration is low (Hülsen *et al.* (2016)). For QASUME and BTS, we assume the correlations to be equally distributed between *full* correlation, *unfavourable* correlation, and *random* correlation (Kärhä *et al.* (2018)). Spectra measured with AVODOR are significantly noisier, thus half of the uncertainty is associated to the *random* component. Values for instability of the calibration lamp are based on long-term monitoring. The lamp irradiances have been noted to gradually drop with no significant wavelength structure within the wavelength region concerned. Non-linearity values are estimations of the operators of the devices. Non-linearity is typically manifested so that the responsivity of the device changes gradually from high readings to low readings. This can cause significant change in the TOC values, thus we assume the correlation type to be

*unfavourable*. Uncertainties due to device stability and temperature dependence are based on long-term monitoring. The changes have been found to be independent on wavelength in the region concerned, thus *full* correlation is assumed. Noise is the average standard deviation of typical measurements at noon over the wavelength region concerned. The wavelength scales of the devices have been checked using emission lines of gas discharge lamps. The uncertainty values given are the estimated standard deviations of the possible remaining errors after corrections. Wavelength error can introduce a significant change in TOC, because it introduces an error in the form of the derivative of the spectral irradiance. Thus, *unfavourable* correlation is assumed. Most of the uncertainty components are slightly wavelength dependent but to simplify simulations, average uncertainty values are used over the wavelength range between 300 nm and 340 nm."

"the discussion about the deviation of the three instruments is very superficial (e.g., "... the reason a the systematic deviation either in the linearity, stray light properties, or the calibration of the device") and inconclusive. How is it connected to the main topic of the paper?"

*Authors:* Thanks to the improved atmospheric model and the improved TOC results, we can remove that sentence. Please, see the first response letter addressed to Referee #2 for full details.

"language inaccuracies are listed in the "Technical corrections" section. One for all: "Izaña results" in the title is very generic. At the Izaña atmospheric observatory (not simply "Izaña"), several activities are organised and the title should appropriately tell which campaign was taken into consideration;"

*Authors:* We change the title of our AMT Discussion manuscript as "Monte Carlo method for determining uncertainty of total ozone derived from direct solar irradiance spectra: The results of ATMOZ field measurement campaign at the Izaña Atmospheric Observatory".

"there is a persistent interchange of terms that should not be mixed: uncertainty, deviation, error, etc. (e.g., "uncertainty induced by deviation", p. 10 line 9);"

*Authors:* In our opinion, we have used those terms correctly, except for "error function" that we change to "deviation function" so that it cannot be confused with "Gauss error function" that is often called simply "error function" or "erf".

Uncertainty is an estimate of the possible error that there may be in a quantity. Spectral errors may further contain systematic deviations. In our analysis we go through systematic deviations (errors) that the uncertainties of the input quantities permit. The variances of TOC obtained with the deviated spectra then give uncertainties for the output quantity TOC. Description of the analysis requires using terms:

- **Standard uncertainty** is the square root of the variance of a probability distribution.
- **Expanded uncertainty** specifies the value of the measurand with 95% confidence. For a Gaussian probability distribution, expanded uncertainty is twice the standard uncertainty ($k$ = 2).
- **Error** is a discrepancy between the measured value and the "true" value.
- Due to the nature of Monte Carlo method, in each Monte Carlo round, we introduce an arbitrary **spectral deviation** in the input parameter and observe how it changes TOC value. When we run Monte Carlo TOC retrieval multiple times, we can calculate the standard uncertainty from the variation in the output TOC.

"QASUME is not only a "high-quality reference instrument … at PMOD/WRC": it is the World reference UV spectroradiometer! Anyway, it should be explained how a global irradiance instrument could measure direct solar irradiance spectra (p. 2 line 5: how was the field of view "limited"?);"

*Authors:* We revise the sentence on page 2, line 1 as:

"One of the instruments is QASUME (Gröbner *et al.* (2005)) that is the World reference UV spectroradiometer at the World Radiation Center (PMOD/WRC)."

We also include a new paragraph about QASUME in Section 2:

"QASUME spectroradiometer collects and guides the incoming radiation with input optics and a quartz fiber bundle to the entrance slit of a Bentham DM150 double monochromator (Gröbner *et al.* (2005)). One wavelength at a time is selected by rotating the two gratings of the double-monochmomator. Then, the monochromatic signal is measured with a photomultiplier tube (PMT). QASUME is usually operated in global spectral irradiance mode (Gröbner *et al.* (2005); Hülsen *et al.* (2016)), but during the campaign it was equipped with a collimator tube with a full opening angle of 2.5° allowing the measurement of direct solar spectral irradiance (Gröbner *et al.* (2017)). The measurement range of QASUME during the campaign was limited to 250 nm – 500 nm with a step interval of 0.25 nm, so that one spectrum was measured every 15 minutes. To ensure stable outdoor measurements, the double-monochromator of QASUME was mounted inside a temperature-controlled weather-proofed box (Hülsen *et al.* (2016))."

**TECHNICAL CORRECTIONS**

"**p. 1 line 2**, "directional irradiance": do you mean "direct irradiance"?"

*Authors:* Yes, we revise the sentence as: "We demonstrate a Monte Carlo model to estimate the uncertainties of total ozone column (TOC), derived from ground-based direct solar spectral irradiance measurements."

"**p. 1 line 2**, "correlations in the spectral irradiance data" is too generic: do you mean correlation of data within the same spectrum and measured at different wavelengths?"

*Authors:* We agree that the sentence is too generic. We revise the sentence as: "The model estimates the effect of possible systematic spectral deviations in the solar irradiance spectra on the uncertainties in TOC retrieved."

"**p. 1 line 20**, "analyse uncertainties": uncertainty of what? I guess in ozone retrievals...;"

*Authors:* Corrected according to reviewer's suggestion. We revise the sentence as: "… data and analyse uncertainties in ozone retrievals for three different spectroradiometers used …"

"**p. 2 line 1-2**: is the order in which the instruments are described (QASUME, AVODOR and BTS) the same as in the abstract (high-end scanning spectroradiometer, high-end array spectroradiometer and roughly adopted instrument)? If not, please avoid confusing the reader;"

*Authors:* We revise the last paragraph in the introduction as:

"In this paper, we introduce a new method for dealing with possible correlations in spectral irradiance data and analyse uncertainties in ozone retrievals for three different

spectroradiometers used in a recent intercomparison campaign at Izaña, Tenerife, to demonstrate how it can be used in practice. One of the instruments is QASUME (Gröbner *et al.* (2005)) that is the World reference UV spectroradiometer at the World Radiation Center (PMOD/WRC). The second one is an array-based high-quality spectroradiometer BTS2048-UV-S-WP (BTS) from Gigahertz-Optik (Zuber *et al.* (2017a, b)), operated by PTB. The third one is an array-based spectroradiometer AvaSpec-ULS2048LTEC (AVODOR) from Avantes, operated by PMOD/WRC. The field of view of the spectroradiometers has been limited so that they measure direct spectral irradiance of the Sun, excluding most of the indirect radiation from the remainder of the sky."

"**p. 2 line 11**: "total ozone content" is mentioned since the beginning of the paper, but formally defined only in page 6 (Eq. 4). Can you move Eq. 4 a bit earlier?"

*Authors:* This is true, and thus we move Eq. (4) of our AMT Discussion manuscript to Section 2 so that now it is the first equation of the manuscript.

"**p. 3 line 1**, "vertical profiles were not implemented": what do you mean?"

*Authors*: We mean that we did not split the atmosphere to horizontal layers at different altitudes and then perform the ozone retrieval fitting. Instead, we use effective atmospheric layers with effective altitudes and temperatures. We revise the sentences in AMT Discussion manuscript as:

"Our ozone retrieval method uses one atmospheric layer to reduce computational complexity. With the one-layer model, the ozone absorption cross-section is a function of the effective temperature, and the relative air mass is a function of the effective altitude of the ozone layer."

"**p. 3 line 2**, "shift the absolute values": values of what? "but should not have an effect": can you justify this hypothesis?"

*Authors:* PMOD/WRC has a version of the code where the atmosphere is split to horizontal layers at different altitudes and piece-wise calculation of the light transmission is performed. Estimation on possible differences in the TOC uncertainty produced by different layer models will be removed.

"**p. 3 line 10**, "the uncertainties ...are standard deviations": standard deviation of what series/samples? Can you mention which kind of measurements were employed?"

*Authors:* We revise the sentence as: "The uncertainties stated for $h_{eff}$ = 26 km ± 0.5 km and $T_{eff}$ = 228 K ± 1 K are standard deviations estimated from the vertical profiles in Fig. 1, measured during the campaign on the days 7, 14, 21 Sept. 2016."

"**p. 3 line 11**, "One of the instruments was the QASUME...": already said;"

*Authors:* We remove this sentence as it was already mentioned.

"**p. 3 line 14**, "every 15 minutes": explain that use of a scanning radiometer involves slower measurements, and why;"

*Authors:* We include a new paragraph in Section 2:

"The data sets measured by three different spectroradiometers were studied in this work. These spectroradiometers use different techniques to measure the spectral distribution of radiation. Monochromator-based spectroradiometers like QASUME, measure one nearly monochromatic wavelength band at a time, and thus measuring the full spectrum is relatively slow. On the other hand, they usually have significantly better stray light properties than array-based spectroradiometers like BTS and AVODOR, which image the full spectrum at once by dispersing the incoming radiation towards a photodiode array."

We also give measurement intervals of BTS and AVODOR.

"**p. 3 line 15**: what is the spectral range of AVODOR? It is legitimate to say that an instrument "has been corrected"?"

*Authors:* Regarding the former comment, we revise the text in Section 2 as:

"AVODOR spectroradiometer has a stationary grating and a back-thinned cooled CCD array detector in a Czerny-Turner configuration. AVODOR measures the spectrum from 200 nm to 540 nm with a step size of 0.14 nm in the UV region. During the ATMOZ campaign, the field of view of AVODOR was limited to 1.5° by a commercial collimator tube used, J1004-SMA by CMS Ing.Dr.Schreder GmbH. The spectral range of AVODOR was limited between 295 nm and 345 nm by a combination of two solar blind filters to reduce stray light from the visible and infrared parts of the solar spectrum. The solar blind filters were mounted between the collimator tube and the fiber entrance of the spectroradiometer. One spectrum was measured every 30 seconds."

Regarding the latter comment, we revise the sentence as:

"To measure direct solar spectrum, BTS was equipped with a collimator tube with a full opening angle of 2.8° designed by PTB, and it uses an internal filter wheel system with 8 filter positions together with a specific measurement routine to reduce stray light."

"**p. 4 Table 1**: is this table useful? The same numbers are repeated in Table 4. Also, provide bibliographic references about how each term in the "Standard uncertainty" column was calculated;"

*Authors:* We move Tables 1, 2, and 3 in the AMT Discussion manuscript to Section 4. We keep Table 1 although some of its data are repeated in Table 4, as its last row states the combined $k = 1$ uncertainty of the spectral measurement. It also makes the describing text clearer when all spectroradiometers are described in similar tables.

We also provide description how standard uncertainties in Tables 1 – 3 are obtained:

"The uncertainties due to radiometric calibration include factors such as the uncertainty of the standard lamp used, and the additional uncertainty due to noise and alignment. QASUME has been validated using various methods, thus the uncertainty due to calibration is low (Hülsen *et al.* (2016)). For QASUME and BTS, we assume the correlations to be equally distributed between *full* correlation, *unfavourable* correlation, and *random* correlation (Kärhä *et al.* (2018)). Spectra measured with AVODOR are significantly noisier, thus half of the uncertainty is associated to the *random* component. Values for instability of the calibration lamp are based on long-term monitoring. The lamp irradiances have been noted to gradually drop with no significant wavelength dependent structure within the wavelength region concerned. Nonlinearity values are estimations of the operators of the devices. Non-linearity is typically manifested so that the responsivity of the device changes gradually from high readings to low readings. This can cause significant change in the TOC values, thus we assume the correlation type to be *unfavourable*. Uncertainties due to device stability and temperature dependence are based on long-term monitoring. The changes have been found to be independent on wavelength in the region concerned, thus *full* correlation is assumed. Noise is the average standard deviation of typical measurements at noon over the wavelength region concerned. The wavelength scales of the devices have been checked using emission lines of gas discharge lamps. The uncertainty values given are the estimated standard deviations of the possible remaining errors after corrections. Wavelength error can introduce a significant change in TOC, because it introduces an error in the form of the derivative of the spectral irradiance. Thus, *unfavourable* correlation is assumed. Most of the uncertainty components are slightly wavelength dependent but to simplify simulations, average uncertainty values are used over the wavelength range between 300 nm and 340 nm."

Regarding the paragraph above, one reference was updated in the AMT Discussion manuscript:

Kärhä P., Vaskuri A., Pulli T., and Ikonen E.: Key comparison CCPR-K1.a as an interlaboratory comparison of correlated color temperature, J. Phys.: Conf. Ser., 972, 012012, 2018. doi:10.1088/1742-6596/972/1/012012

To clarify the uncertainty components in Table 4, we also include new sentences in the AMT Discussion manuscript before line 5 on page 14:

"For components (a) – (d) in Table 4, the mechanism of contributing to the uncertainty of TOC is known. We know the standard uncertainty of the $O_3$ layer altitude of 26 km to be $u$ = 0.5 km, so we vary the altitude accordingly and note the variance of the resulting TOC."

"**p. 5 line 2**, "fitting the ozone retrieval": can a single quantity be fitted?"

*Authors:* We revise the sentence as: "They are needed when fitting the spectra at the Earth surface modelled with the ozone retrieval to the measured spectra."

"**p. 5 line 5**, "affects uncertainties with a factor of sqrt(N)": do you mean 1/sqrt(N)? If so, why the Brewer - which measures the irradiance for the ozone retrieval at only 4 wavelengths - is considered a reference in the paper?"

*Authors:* We revise the sentence as: "In our full spectrum TOC retrieval, the number of data points $N$ which is smaller with a larger wavelength step interval, affects uncertainties with a factor of $1/\sqrt{N}$ (Kärhä *et al.* (2017b); Poikonen *et al.* (2009))."

We include a new reference in the AMT Discussion manuscript:

Poikonen T., Kärhä P., Manninen P., Manoocheri F., and Ikonen E.: Uncertainty analysis of photometer quality factor $f_1'$, Metrologia, 46, 75–80, 2009.

Regarding the latter question, our full spectrum retrieval method does "averaging" in the wavelength domain, whereas Brewer spectrophotometer does it in the time domain. Brewer can measure up to tens of seconds to get millions of photons, so that the photon noise reduces to a level of 0.1%. At this low noise level, it is not critical that only four wavelengths

are used. Averaging over multiple measurement sequences, regardless of the retrieval method used, reduces the noise in TOC by a factor of $1/\sqrt{N_r}$, where $N_r$ is the number of repetitions.

"**p. 6 line 1-2**: give credit to Bouguer, Lambert and Beer (not Huber et al. 1995);"

*Authors:* We include references to Bouguer, Lambert, and Beer. We also have to acknowledge Huber *et al.*, as the complete ozone retrieval using spectral irradiance measurements and least-squares fitting method have been documented in that paper, and it is also one of the most useful references of our manuscript. Thus, we revised the beginning of Section 3 as:

"In this study, we use an atmospheric ozone retrieval algorithm in many aspects similar to the article by Huber *et al.* (1995). The relationship between the spectral irradiance $E_s(\lambda)$ at the Earth surface and the extra-terrestrial solar spectrum $E_{ext}(\lambda)$ is based on Beer-Lambert-Bouguer absorption law (Beer (1852); Lambert (1760); Bouguer (1729)) as ..."

New references are included in the AMT Discussion manuscript:

Beer A.: Bestimmung der Absorption des rothen Lichts in farbigen Flüssigkeiten, Annalen der Physik und Chemie., 86, 78–88, 1852.

Bouguer P.: Essai d'optique sur la gradation de la lumière (Paris, France: Claude Jombert, 1729) pp. 16–22.

Lambert J. H.: Photometria sive de mensura et gradibus luminis, colorum et umbrae (Augsburg ("Augusta Vindelicorum"), Germany: Eberhardt Klett, 1760).

"**p. 6 Eq. 3**: a reference to the used extraterrestrial spectrum (QASUME-FTS) should be mentioned just after the equation;"

*Authors:* Corrected according to reviewer's suggestion. We include a new sentence in our AMT Discussion manuscript after line 6 on page 6:

"The QASUME-FTS data set by Gröbner *et al.* (2017) was used as the extra-terrestrial spectrum $E_{ext}(\lambda)$."

"**p. 6 line 12**: theta is the angle at the observing site, not the angle between vacuum-to air interface;"

*Authors:* Yes, it is the angle at the observing site. Fortunately, this was a mistake only in the manuscript. It was correctly used in the code. First, effect of the solar zenith angle $\theta_v$ at the vacuum-to-air interface at the effective altitude $h_{eff}$ on the relative air mass $m$ is defined as

$$m = \frac{1}{\cos(\theta_v)}. \tag{1R}$$

After taking the Earth curvature into account, we obtain the relative air mass dependence on the solar zenith angle $\theta$ at the observing site as

$$m = \frac{1}{\cos\left[\arcsin\left(\frac{R}{R+h_{eff}}\cdot\sin\theta\right)\right]}, \tag{2R}$$

where $R$ is the Earth radius.

"**p. 7 line 10**: the extinction coefficient is defined as dTau/dz, thus it has nothing to do with beta;"

*Authors:* Corrected according to reviewer's suggestion. Now, we call $\beta$ as the Ångström turbidity coefficient.

"**p. 7 line 11**: avoid the expression "terrestrial spectrum", the radiation is from the sun, not from the Earth. Use "solar spectrum at the Earth surface";"

*Authors:* Corrected according to reviewer's suggestion.

"**p. 7 line 22**, "As can be seen, the signal-to-noise ratios ... differ": how can I see it from the figure, without any explanation?"

*Authors:* It is true that the baseline noise was not clearly shown in Fig. 3 in our AMT Discussion manuscript due to the scaling of the axes. We replace Fig. 3 with Fig. 1R below and revise the paragraph referring to this figure. The new paragraph reads:

*"Figure 3 presents examples of measurements and modelled values for the spectroradiometers used in this work. As can be seen, the signal-to-noise ratios of the devices differ significantly among different spectroradiometers. All spectra measured by QASUME spectroradiometer are excellent above $10^{-6}\,\mathrm{Wm^{-2}nm^{-1}}$ with the dynamic range of approximately four orders of magnitude. The dynamic range for BTS is approximately two orders of magnitude and for AVODOR it is less than two orders of magnitude. Based on the analysis in Appendix A, we use absolute least squares minimisation in TOC estimation with $w(\lambda) = 1$ for BTS and AVODOR as it is not affected by the lowest irradiance levels where the stray light and noise are dominant. For QASUME, we use the relative least squares minimisation with $w(\lambda) = E(\lambda)^{-2}$ as it has been used in the past for monochromator-based spectroradiometers, e.g. by Huber et al. (1995). The shortest modelling wavelength for the spectroradiometers in this work was set to 300 nm since the typical stray light corrections are not effective below 300 nm (Nevas et al. (2014)). The upper wavelength limit was set to 340 nm with all three spectroradiometers as the ozone absorption is not effective above that wavelength."*

[Figure]

*Figure 1R. Examples of fitting the atmospheric model to the direct ground-based solar UV spectra between 300 nm and 340 nm for QASUME (a–b), BTS (c–d) and AVODOR (e–f). In figures on the left hand side, the coloured symbols indicate measured spectra, and the black solid curves indicate modelled spectra. Figures on the right hand side show the relative spectral residuals of the fits. In (a), the abbreviation DR refers to the dynamic range of QASUME data used in the least squares fitting.*

"**p. 9 line 3**: "noise" usually defines a random variable, while stray light is a systematic effect. Don't put them together in the same sentence;"

*Authors:* The new data analysis using absolute least squares fitting for AVODOR and BTS makes this sentence obsolete. The whole paragraph will be rewritten.

Second response letter to Referee #2
Article reference: amt-2017-403

"**p. 10, Eq. 11**: define "u";"

>    *Authors:* We include a following text after line 5 on page 10: "… where $u(\lambda)$ is the relative standard uncertainty of the spectrum."

"**p. 10 line 20**, "does not have any internal limitation to the shape of the error function": what do you mean?"

>    *Authors:* The MC uncertainty model does not assume any particular spectral shape of the deviation. Fourier series has a property that in its full form it can produce any shape of deviation. This takes place at the Nyquist criterion, where $N$ is equal to half the number of wavelengths available. Also with smaller values of $N$, the MC parameters can account for unknown spectral shapes of lower complexity.

"**p. 11 line 3**, "components stating fractions": what do you mean?"

>    *Authors:* We revise the sentence as: "The uncertainty components divided to the three correlation types have been analysed with the new model. The other components (a)–(d) have been solved using traditional MC modelling because the mechanism for the uncertainty propagating to TOC is known."

"**p. 12 Table 4**: why is "X" used instead of "TOC"?"

>    *Authors:* We replace "$\tau_X m_X$" with "in exponent" in Table 4 to avoid confusion.

"**p. 12 line 2**: was "r" defined?"

>    *Authors:* Yes, as there is a sentence before Eq. (12): "Division of the correlation to the three categories introduced are stated for each row as fractions $r_{full}$, $r_{unfav}$, and $r_{random}$."

"**p. 13 line 20**, "a wavelength shift will introduce unfavourable correlations": why? The ozone cross section has a complex shape;"

>    *Authors:* The ozone absorption cross-section has a complicated shape and the spectral deviation due to the wavelength shift is then also complicated. In other words, the uncertainty due to this complex correlation causes higher uncertainty than noise could cause. Thus, we assume *unfavourable* case of correlation.

"**p. 13 lines 24-25**, "the wavelength shift... should be corrected for the extraterrestrial spectrum": or vice-versa?"

>    *Authors:* We revised the sentence as "the wavelength shift ... should be corrected **from** the extra-terrestrial spectrum".

Second response letter to Referee #2
Article reference: amt-2017-403